# THE VALUE OF SENSORY INFORMATION TO A ROBOT

**Arjun Krishna**
University of Pennsylvania
arjk@seas.upenn.edu

**Edward S Hu**
University of Pennsylvania

**Dinesh Jayaraman**
University of Pennsylvania

## ABSTRACT

A decision-making agent, such as a robot, must observe and react to any new task-relevant information that becomes available from its environment. We seek to study a fundamental scientific question: what value does sensory information hold to an agent at various moments in time during the execution of a task? Towards this, we empirically study agents of varying architectures, generated with varying policy synthesis approaches (imitation, RL, model-based control), on diverse robotics tasks. For each robotic agent, we characterize its regret in terms of performance degradation when state observations are withheld from it at various task states for varying lengths of time. We find that sensory information is surprisingly rarely task-critical in many commonly studied task setups. Task characteristics such as stochastic dynamics largely dictate the value of sensory information for a well-trained robot; policy architectures such as planning vs. reactive control generate more nuanced second-order effects. Further, sensing efficiency is curiously correlated with task proficiency: in particular, fully trained high-performing agents are more robust to sensor loss than novice agents early in their training. Overall, our findings characterize the tradeoffs between sensory information and task performance in practical sequential decision making tasks, and pave the way towards the design of more resource-efficient decision-making agents. Appendices, videos, and more at https://sites.google.com/view/vosi-robotics/

## 1 INTRODUCTION

Over time, as a robot moves, information emerges in its environment (e.g. an occlusion is removed and a new object becomes visible, or a pair of dice settles on an outcome), some of it is captured by the robot's sensors, and the robot then acts on this information. Any information not acquired at sensing is lost to all downstream decision-making computations, and perhaps for this reason, it is often taken for granted during the design of a robot that *more sensing* always improves task performance. However, sensing is not always an asset (Mason, 1993; Erdmann & Mason, 1988). Sensing and the processing of sensed information entail the use of several key resources in robotics: computation latency, power, controller complexity, and by extension for *learning* robots, training data. Even discounting these resource costs, over-sensing can still hurt task performance by causing fragile robot behaviors that respond to task-irrelevant sensed details in the environment (Mason, 1993).

Understanding the relationship between the sensing setup (how much information is sensed, what is sensed, quality, timing, latency etc.) and the performance of a downstream policy is an important scientific problem in robotics and more broadly, all of sequential decision-making (Donald, 1995; LaValle, 2019; Tishby & Polani, 2010; Koditschek, 2021; Xu et al., 2021; Majumdar et al., 2023). In this paper, we first propose a novel approach to empirically study one slice of this question: at what instances along the robot's trajectory does environment feedback reveal valuable information? In particular, we assume a fixed policy and a simulated perfect sensor that instantaneously reveals the full Markov state of the environment. In this setting, our approach (Section 5) characterizes: at what moments did access to the sensed information critically improve the robot's actions? We study several benchmark tasks spanning varying robot morphologies, and diverse policy synthesis approaches, particularly focusing on state-of-the-art imitation and reinforcement learning techniques. Our analysis yields several interesting findings, of which we highlight a select few here. In most tasks, sensing only improves agent actions significantly at some rare moments during task execution. Further, the reliance on environment feedback is similar for high-performing policies synthesized

through very different techniques and represented with very different architectures. Finally, and rather counter-intuitively, policy performance is inversely correlated with sensor dependency. Well-trained policies respond to sensed information much less frequently. We summarize our contributions below:

- We propose a novel approach to quantify the value of sensory information (VoSI) in complex task setups focusing on environments with perfect sensing and deterministic dynamics.
- With VoSI as a probing tool, we systematically analyze diverse robotics-inspired tasks and state-of-the-art lookahead policies, revealing novel insights into the role of sensing.
- While these robotic settings largely feature deterministic dynamics and perfect sensing, we also validate VoSI in more toy tasks with stochastic dynamics and model or sensing noise.
- We highlight the potential of VoSI for enabling efficient policy execution by demonstrating how to construct a policy that only senses at key sensing moments.

## 2 RELATED WORK

**Information Parsimony through Constrained Optimization.**    Agents in the real world often incur significant costs in acquiring and processing sensory information. The trade-off between maximizing task-performance and minimizing associated sensing costs has motivated extensive work on synthesizing efficient controllers under resource constraints. Hansen et al. (1996) ascribe a fixed cost to any sensing action and learn to optimize when to sense along with maximizing a task objective in discrete MDPs. Recently, Treven et al. (2024); Holt et al. (2024) associate constraints with the number of interactions in continuous time control settings. Tishby & Polani (2010); Eysenbach et al. (2021); Lu et al. (2023) propose associating a cost with the bits of information inferred from observations without any direct constraints on sensing itself. In control theory, the event-triggered formalism (Aström, 2008) enables the design of controllers that process full-state information only when some coarsely sensed deviations are observed, thereby reducing the computational overhead. These works set up specific models for the resource costs entailed by sensing, and aim to generate good task policies under constraints on those costs. Complementing these approaches, we instead build a general tool to understand the sensory requirements for a fixed task performance behavior, as represented in a frozen controller. We do this by setting up and measuring the "value of sensory information". Our notions of task value are pertinent for efficient execution of that behavior under any resource cost models and constraints, when the robot looks to act efficiently. We show that our approach reveals interesting insights into the properties of lookahead-policies on challenging tasks.

**Value of Information (VoI) For Sequential Decision Making Agents.**    Howard (1966) first described the "value of information" (VoI) towards decision making, placing a value on the reduction of uncertainty about various random variables relevant to a task. Inspired by this highly influential framework, we propose to measure the "value of sensory information" for a sequential decision making agent, to assign values to sensory measurements made at various moments in time. We are not, however, the first to apply VoI-related ideas in control. Notably, Flaspohler et al. (2020) build on VoI to synthesize near-optimal model-based planners for PoMDPs: they construct open loop "macro actions" that significantly reduce the planning complexity. Their macro-actions represent a restriction of the mixed loop policy executions that we study. Further, their experimental validation is restricted to stylized simple environments. Majumdar et al. (2023) characterize the task-relevant information potential of sensed information and provide fundamental bounds on the performance achievable by any sensor-based policy. This characterization requires environments that can be expressed in simple analytic forms or that are tractable for sampling based assessments (requiring finite actions and short task horizon), and this is reflected in their experiments. Further, they do not offer insights on *when to sense*. In this work, we propose a novel empirical analysis approach that can be applied to much more complex environments and policy representations than these prior works. Through it, we study state-of-the-art robotic policy architectures, on standard benchmark tasks in the robot learning literature, revealing insights about the value of sensory information to an agent.

## 3 SETUP AND NOTATION

We model the environment as a controlled Markov Process $(S, A, \mu, \mathcal{P})$, where $S$ is the set of states $s$, $A$ is the set of control actions $a$ available to a robot, $\mu$ is the distribution of start states, and

$$s_0 \xrightarrow{a_0} P(s_1|s_0, a_0) \xrightarrow{a_1} P(s_1|s_0, a_0)P(s_2|s_1, a_1) \xrightarrow{a_2} s_3 \xrightarrow{a_3} P(s_4|s_3, a_3) \xrightarrow{a_4} s_5$$

$$[a_0, a_1, a_2] \sim \pi_{0:3}(s_0) \qquad\qquad [a_3, a_4] \sim \pi_{0:2}(s_3)$$

Figure 1: Illustration of a mixed loop (MixL) execution of a look-ahead policy $\pi$. At state $s_0$, the agent executes the open loop action plan $a_{0:3} \sim \pi_{0:3}(s_0)$ for the next $h = 3$ steps. Over this open loop execution, uncertainty accrues due to stochastic dynamics and/or an imperfect environment model, that is then resolved when the agent senses the state at $t = 3$.

$\mathcal{P}(s_{t+1}|s_t, a_t)$ is the transition probability distribution specifying how the environment state $s_t$ at time $t$ evolves in response to a robot action decision $a_t$. We consider two common modes of "task specification". The first is imitation learning, where the agent must mimic expert actions closely. The second is reinforcement learning, where a reward function $r_t = r(s_t, a_t)$ specifies feedback, and the agent must maximize (undiscounted) cumulative rewards $\mathbb{E}_{\mathcal{P}}[\sum_{t=0}^{T} r(s_t, a_t)]$.

**Look-Ahead Policies.** For our analysis, we consider the popular class of "look-ahead" policies $\pi : S \to A^n$ that, after observing the current state $s_t$, prescribe not just the immediate action $a_t$, but an $n$-step "chunk" of future actions $\pi(a_{t:t+n}|s_t)$. This is a common feature in recently popular imitation learning architectures such as diffusion policy (Chi et al., 2023), as well as popular planning-based approaches such as the model-based RL architecture TD-MPC2 (Hansen et al., 2023).

**Mixed Loop and Fixed-Rate Execution.** Crucially for our analysis, such look-ahead policies permit execution in a "mixed loop mode". At any given state $s_t$, the agent can commit to executing any $h \leq n$ steps of the action chunk $a_{t:t+h}$ in an open loop manner. At time $t + h$, the loop is closed by sensing and reacting to $s_{t+h}$ (see Figure 1). Fixing $h = 1$ corresponds to closed loop execution. Popular look-ahead policies often reduce resource usage for on-robot execution by sensing and acting periodically with fixed "execution period" $h > 1$ to reduce resource usage (Chi et al., 2023). In other words, the agent forgoes sensory information between times $t$ and $t + h$.[1] We call this "fixed-rate" mixed loop execution. More generally, $h < n$ need not be fixed. We denote the distribution of actions for $h$ timesteps conditioned on observing $s_t$ as $a_{t:t+h} \sim \pi_{0:h}(\cdot|s_t)$.

## 4 HOW FREQUENTLY DO AGENTS NEED TO SENSE?

Our efforts to study the value of sensory information start by envisioning a robotic agent that is provided with a look-ahead policy $\pi : S \to A^n$, synthesized by any means of our choice e.g., model-based control, imitation learning, or reinforcement learning. It must then decide "how to execute" this policy starting from some state $s_0$, in particular, whether to "sense" or "not" at each time instant $t = 1, 2, \ldots$. As motivated in the introduction, each "sense" decision carries resource costs to real agents. However, for most of our investigation, rather than set specific resource costs, we instead study how much positive value in terms of task performance improvements each instant of sensory information brings to the agent to offset any potential resource costs.

Mixed loop execution with lookahead policies enables us to empirically study the impact of forgoing sensing at any time(s) on the performance of the policy $\pi$. In the general mixed loop setting, for a time duration $T$ starting at a sensed state $s_0$, there are $2^T$ possible executions, including fully closed loop (sense at each moment) and fully open loop (never sense), amongst which our agent must choose. For this first experiment, we restrict our attention to the simpler class of fixed-rate mixed loop executions with varying execution period $h$, to arrive at an approximate understanding of the relationship between the sensing budget (as reflected in the sensing rate $1/h$), and task performance.

We study 7 diverse robotic tasks, depicted in Figure 2, sourced from DM-control (Tassa et al., 2018), Robosuite (Zhu et al., 2020; Mandlekar et al., 2023), and the Push-T task (adopted from Chi et al. (2023)). The DM-Control tasks involve dynamic behaviors with varied robots and action spaces, like swinging up a cartpole, catching a ball in an actuated cup, and spinning a wheel at high angular

---

[1] The policy "action" is often issued to a low-level controller such as PID that uses proprioceptive feedback.

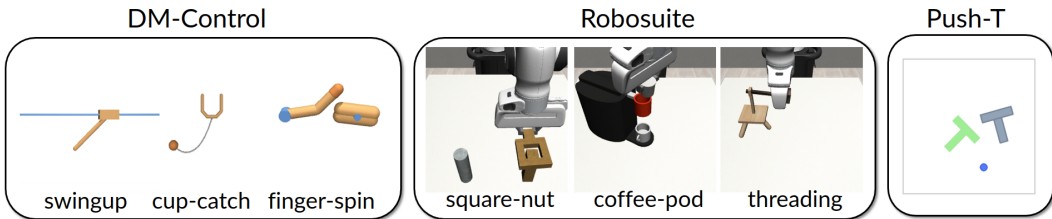

Figure 2: The seven robotic tasks considered in our analyses.

velocity with an actuated 2-jointed finger. Robosuite is a popular robot learning benchmark for static tabletop object manipulation tasks: we study `square-nut`: picking up and placing a square nut around a square peg, `coffee-pod`: placing a coffee pod into a coffee machine slot, and `threading`: a high-precision task involving threading a thin needle through a small hole on a light movable platform. Finally, `Push-T` is a stylized 2D non-prehensile manipulation task that involves complex contact dynamics between a cylindrical end-effector (projected to a circle) and a T-shaped object that must be pushed into a target configuration.

For each task, we train state-based lookahead-policies with TD-MPC2 (Hansen et al., 2023) and DiffusionPolicy (DP) (Chi et al., 2023), which represent the state of the art in model-based RL and imitation respectively. TD-MPC2 failed to produce effective policies on the Robosuite manipulation tasks, and DP failed on finger-spin and cup-catch tasks. We study the other policies, which all performed well. More details of training setup and data-collection are provided in Appendix A.

Figure 3 plots the best mean reward with 95% confidence intervals i.e. lowest regret achieved by fixed-rate mixed loop executions as the budget varies. "Normalized regret" is measured as a fraction of the mean closed loop execution reward, which is always positive in our tasks. There are several outstanding trends: most task policies in our study perform just as well as closed loop ($\approx 0$ normalized regret), even when executed once every 5 steps. The three Robosuite tasks are remarkable: for all these manipulation tasks, policy performance does not deteriorate substantially even when sensing happens only once, at the beginning of each trajectory. We refer to such "one-time-sensing" execution as "open loop" in this paper. This surprising effectiveness of open loop executions is in line with observations made in recent works Dasari et al. (2022); Raffin et al. (2023); Wang et al. (2024) for other standard robot learning benchmark tasks. Two tasks DM-control `swingup` and `Push-T` afford comparing TD-MPC2 and DP policies. Trends are largely similar for the two: on `Push-T`, TD-MPC2 appears to be slightly more robust (lower regret) to reduced sensing. Finally, `cup-catch`, `finger-spin` and `Push-T` all show smoothly accelerating regret as the sensing rate decreases.

**A Note On Regret Profiles and Task Complexities.** Each task's regret profile may also be interpreted as a measure of its "sensory complexity": low regrets indicate low sensory complexity. By this measure, the tasks in order of difficulty are: `coffee-pod` $\approx$ `square-nut` $\approx$ `threading` < `swingup` < `cup-catch` < `finger-spin` $\approx$ `Push-T`. There is thus also quite a clear complexity ordering among task suites: Robosuite < DM-control < Push-T. Speculating a little, robot learning benchmarks for manipulation as represented by these three Robosuite tasks may be outliers in terms of how little sensing / perceptual capabilities they require.

## 4.1 WHICH CONDITIONS NECESSITATE SENSING?

To appropriately interpret these task regret profiles, we now analyze how task characteristics and policy architectures drive policy performance under reduced sensing.

**The Sensing Needs of Optimal Infinitely Expressive Policies.** Sensing has value to an agent to the extent that it delivers information that the agent does not already know, or more explicitly, what the agent could not have predicted without sensing. Let us consider how a particular observation $s_t$ can be informative to an *optimal* policy $\pi^*$, without limitations on its expressivity or capacity. Observe that we can construct one such optimal $\pi^*$ that operates by optimizing future actions $a_{t:T}$ through the true environment transition function $P(s'|s, a)$, either learned or manually specified. Now, if an agent operating with $\pi^*$ knows the current state $s_t$, then its uncertainty about $s_{t+1}$ is the entropy of the

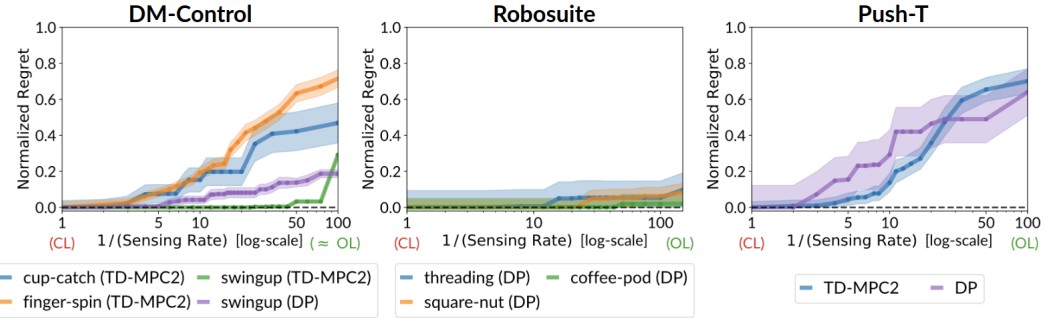

Figure 3: **Task regret profiles** on our 7 main tasks. Plots demonstrating what fraction of closed loop policy returns are lost ("normalized regret") when a policy is instead run in fixed-rate mixed loop mode. We group the tasks into DM-control (**left**), Robosuite (**middle**), and Push-T (**right**).

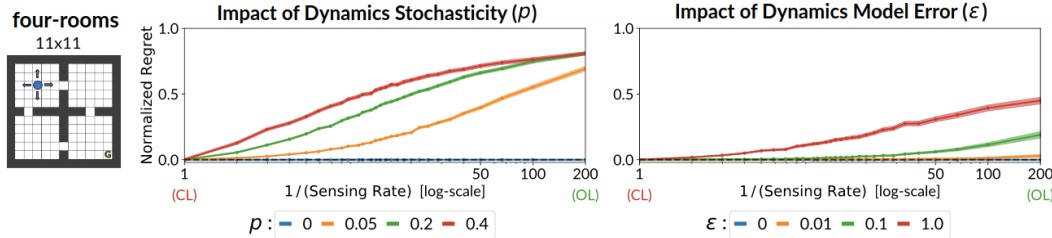

Figure 4: Task regret profiles on our toy `four-rooms` task (**left**) illustrate how stochastic dynamics (**middle**) and dynamics model error (**right**) necessitate more sensing.

transition distribution: $H(s_{t+1}) = -\sum_{s'} P(s'|s_t, a_t) \log P(s'|s_t, a_t)$. In other words, sensing $s_{t+1}$ delivers $H(s_{t+1})$ bits of information. It follows that the *task-relevant* information value of sensing $s_{t+1}$ is bounded as $H_{tr}(s_{t+1}) < H(s_{t+1})$. If the dynamics $P(s'|s, a)$ are always deterministic, then $H(s_{t+1}) = 0$: there is no value to sensing $s_{t+1}$, or by induction, any future states. This is intuitive: $\pi^*$ can precisely simulate $s_{t+1}$: and can therefore act optimally without ever needing to actually sense the environment.

Thus, there is at least one task-optimal policy $\pi^*$ that can operate "open loop" (i.e., sensing only at $t = 0$) when sensing is perfect and dynamics are deterministic. A corollary of this result is that if the task reward includes any non-zero cost $c > 0$ associated with sensing, then $\pi^*$ *must* be open loop.

To illustrate, consider a simple 11x11 `four-rooms` discrete grid-world task where a robot is tasked to reach and stay at a fixed goal location starting from anywhere (see Figure 4). The robot has one stay-in-place action and 4 directional actions, and gets a sparse reward of 1 when it reaches the goal. We assume perfect sensing in all environments in this work, but we now introduce stochastic dynamics in `four-rooms`: an action other than the desired command $a$ is executed at random with probability $p$. We obtain a task-optimal model-based agent ($\pi^*, \hat{P}^* = P$) for each environment, and compute similar sensory-regret profiles for fixed-rate mixed loop executors of agents across different levels of stochasticity ($p$). Observe that an optimal agent gains no additional value from sensing after the first timestep under deterministic environment dynamics ($p = 0$) and that the value gain from sensing (performance regret) is correlated with the level of stochasticity.

**How Finite Expressivity Influences Sensing Needs: Complexity $\leftrightarrow$ Uncertainty.** Our 7 simulated robotic tasks above have perfect sensing and deterministic dynamics, as in most robotic benchmark tasks. Why then do agents still perform worse when they can't sense and react to environment states? We argue that for "finite policies" (e.g. finite in expressivity or training data), *complexity* plays a similar role to stochasticity in the above analysis. Intuitively, finite policies struggle to simulate complex dynamics, such as of a ball bouncing off a rocky forest floor, or of a swimmer swimming through a perturbed river stream. This also applies to complexity in sensing or actuation: provided with a perfect high-resolution 3-D map of the forest floor or a flow field of the river stream, finite policies may not successfully process such observations into the task-relevant state. To see this in

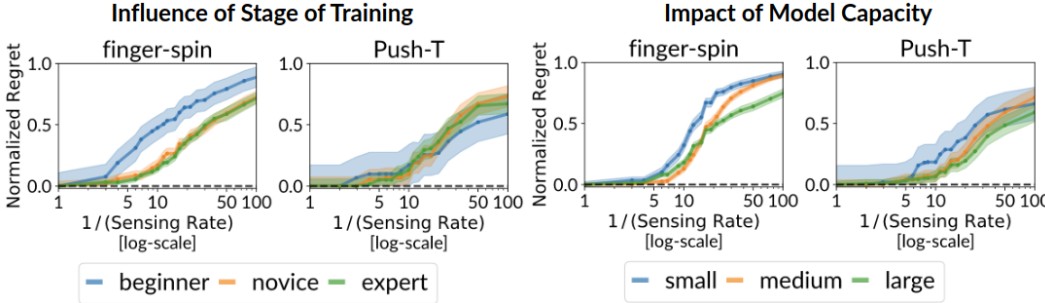

Figure 5: Task regret profiles on two tasks, showing how increasing policy competency affects sensing needs, through **(left)** various training stages , and **(right)** various model capacities.

our grid-world family, consider agents on a deterministic grid, but with different levels of model error $\epsilon$, such that $D_{\mathrm{KL}}(P, \hat{P}) = \epsilon \, \forall (s, a)$, simulating the implications of finite expressivity (see Appendix B for more details on how $\hat{P}$ is generated). As Figure 4 shows, these model errors generate similar regret profiles to true environment stochasticity above. On the robotic benchmark tasks `finger-spin` and `Push-T` we explore the implications of finite training data and model-capacity by evaluating the task regret profiles of fixed-rate mixed loop executors of TD-MPC2 agent checkpoints at different stages of training: early (beginner), middle (novice), and latest (expert) and different representation capacities (modulated by the size of the latent dimensions of the MLP) in Figure 5 (details of the experiment design in Appendix A.1.1). In line with the arguments above, agents with higher model capacity or those exposed to more training data appear to degrade less at lower sensing rates. Note that the relative drop in performance requires some interpretation in the context of the reward function and absolute returns attained – we discuss this in Appendix F.

## 5 WHEN SHOULD AN AGENT SENSE?

We have interpreted the regret plots of Figure 3 as coarsely profiling the importance of sensory information in a task. However, those experiments only considered fixed-rate mixed loop policies and studied the impact of a fixed sensing frequency on task performance. They do not support finer-grained questions about the *moments in time* at which sensing is more valuable or less valuable. For example, in the push-T task, where contact dynamics only come into play when the end-effector is near the T, states at the beginning of an episode as the end-effector moves to the T might not require closed loop execution (as we will show later in Figure 8). We now build on from the methodology of Section 4 to permit such detailed analysis.

**The Value of Sensory Information (VoSI).** Having sensed the state $s_t$ of the world at time $t$, we seek to understand how much *task-relevant value* an agent loses by choosing to not sense the state of the world for $h$ timesteps compared to operating closed loop. To empirically characterize this state-wise regret, let us first consider $\eta_{\mathrm{MixL}}(s_t, h)$, a mixed loop "execution strategy" that observes state $s_t$ at time $t$, then forgoes sensing for the next $h$ steps, and operates closed loop forever after. This strategy $\eta_{\mathrm{MixL}}(s_t, h)$ induces a distribution over trajectories $\tau_{\mathrm{MixL}}(s_t, h)$ of episode length $T$. For convenience, we denote the distribution induced by closed loop control ($h = 0$) as

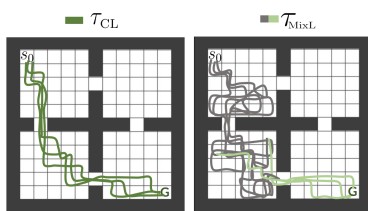

Figure 6: $\tau_{\mathrm{MixL}}(s_0, h)$ and $\tau_{\mathrm{CL}}(s_0)$ for `four-rooms`.

$\tau_{\mathrm{CL}}(s_t) := \tau_{\mathrm{MixL}}(s_t, 0)$. Some $\tau_{\mathrm{CL}}$ and $\tau_{\mathrm{MixL}}$ trajectories are visualized in Figure 6 for `four-rooms` with stochastic dynamics. The open loop phase of $\tau_{\mathrm{MixL}}$ trajectories are depicted in gray, and the closed loop phase that follows, in green.

The idea is simple: $\tau_{\mathrm{MixL}}$ represents the best behaviors when sensing is withheld over the window $(t, t + h]$, and $\tau_{\mathrm{CL}}$ in turn represents the best behaviors without any sensing restrictions. Thus, any deterioration in task performance from $\tau_{\mathrm{CL}}$ to $\tau_{\mathrm{MixL}}$ may be attributed to task-relevant sensory information lost during $(t, t + h]$, and a measure of that deterioration also serves to quantify the "value of sensory information" (VoSI) in that window. When a reward function is available, as in our

tasks, an obvious choice of a task-relevant deterioration measure is the loss of rewards of the mixed loop execution relative to a closed loop execution, until the end of the episode. Thus the VoSI is:

$$\text{VoSI}(s, h) := \mathop{\mathbb{E}}_{\tau_{\text{CL}}(s)} \left[ \sum_{t=0}^{T} r_t \right] - \mathop{\mathbb{E}}_{\tau_{\text{MixL}}(s,h)} \left[ \sum_{t=0}^{T} r_t \right]. \tag{1}$$

This is closely related to the task-level regret profiles analysis of Section 4. When measured for a fixed state $s$, with increasing $h$, $\text{VoSI}(s, h = [1, 2, ...])$ represents a kind of state-wise regret profile: it measures how task performance degrades when acting open loop starting from $s$ for increasing durations of time. We call these **VoSI profiles**. Later, we will define and compare other alternative measures of deterioration from $\tau_{\text{CL}}$ to $\tau_{\text{MixL}}$, to plug into VoSI computation in specific settings, such as when no reward is available.

Before moving on to further experiments using this measure of the value of sensory information, a note: recall from the beginning of Section 4 that our study is limited to an agent considering various "executions" of a frozen look-ahead policy $\pi$, distinguished only by the moments in time when it is afforded access to sensory inputs $s_t$. One practical issue in this evaluation is that the standard pre-trained look-ahead policies $\pi$ (such as TD-MPC2, DiffusionPolicy) are designed for closed loop performance, and would be operating out-of-distribution when executed in mixed loop mode. Thus $\tau_{\text{MixL}}$ as defined above might produce erratic behaviors, and might not produce the best execution of $\pi$ subject to the constraint of not sensing within a time window. However, in our experiments, mixed loop $\tau_{\text{MixL}}$ executions of these policies appear to hold up well enough to produce coherent and interpretable findings, and those findings themselves align with the expectations we arrive at through reasoning as in Section 4.1.

**Implementing VoSI Profiles.**  We measure VoSI on all 7 tasks from Figure 2, for TD-MPC2 (DM-control, Push-T) and DP policies (Robosuite). In all cases, we first execute the policies closed loop starting from the task's initial state distribution, and then measure VoSI on states $s$ from these closed loop trajectories as follows. For each state $s$, we first reset the simulator to that state, and then run mixed loop strategies $\eta_{\text{MixL}}(s, h)$ with the open loop length $h$ varying from 1 to 100 steps. For each $h$, we generate 5 full trajectories starting at $s$ and ending at episode termination (after $T$ steps). This permits a Monte Carlo estimate of the expectation over $\tau_{\text{MixL}}(s, h)$ in the second term in Equation (1). For the expectation over $\tau_{\text{CL}}$ in the first term, we similarly use 5 trajectory rollouts to estimate this quantity. These are expensive evaluations: for each state for which we compute VoSI, we must generate 100x5 = 500 trajectories. For DP which is computationally expensive to run, we restrict the analysis to interesting states, ignoring states where the arm is primarily moving in free space (details in Appendix A.3).

**The Informative Shapes Of VoSI Profiles.**  Figure 7 presents a few representative VoSI profiles $\text{VoSI}(s, h = [1, 2, ...])$ from across all tasks, sorted into three prototypical profile "shapes".

▶ **"Flat"** VoSI profiles (Figure 7-1) occur when the agent derives very little value from sensing in $(t, t + h]$ even at high $h$. In other words, having sensed the state at $t$, the agent can operate without sensing, losing nearly no task reward. In Robosuite, as suggested earlier by the task regret profiles of Figure 3, VoSI profiles are surprisingly flat even during object interactions (see e.g. Figure 7-1g). On DM-control and Push-T, VoSI profiles usually only become "flat" after the task goal is effectively achieved (e.g. the T is already in place in Push-T, see Figure 7-1e), and the agent does not need to do much other than wait for episode termination. For swingup, VoSI profiles are flat both near the starting states where the dynamic swing action must be initiated (Figure 7-1b), and more surprisingly, even in the upright phase (Figure 7-1c) up to about 70 steps ($\approx 0.7$ seconds). Figure 8-1 offers some insight on this latter case: closed and open loop trajectories ($\tau_{\text{CL}}$ and $\tau_{\text{MixL}}$) look remarkably alike, because good policies barely actuate the cartpole base in the upright phase.

▶ **"Gradual"** VoSI profiles (Figure 7-2) involve steadily accruing VoSI as the window length $h$ increases, suggesting that the task-relevant value of sensing grows steadily with the delay in acquiring new information. Sustained and repetitive contacts as in Push-T and finger-spin (Figure 7-2a-e) often induce this VoSI prototype. This is understandable: contact is hard to model precisely, so the true realizations of contact dynamics as represented in sensed states always resolve some ambiguity when available. Figure 8-2 zooms into the Push-T state in (Figure 7-2e): over the course of a push, small missteps from open loop execution cause steady drift from optimal trajectories. This requires extra recovery time when closed loop control resumes after $h$ steps, affecting task reward.

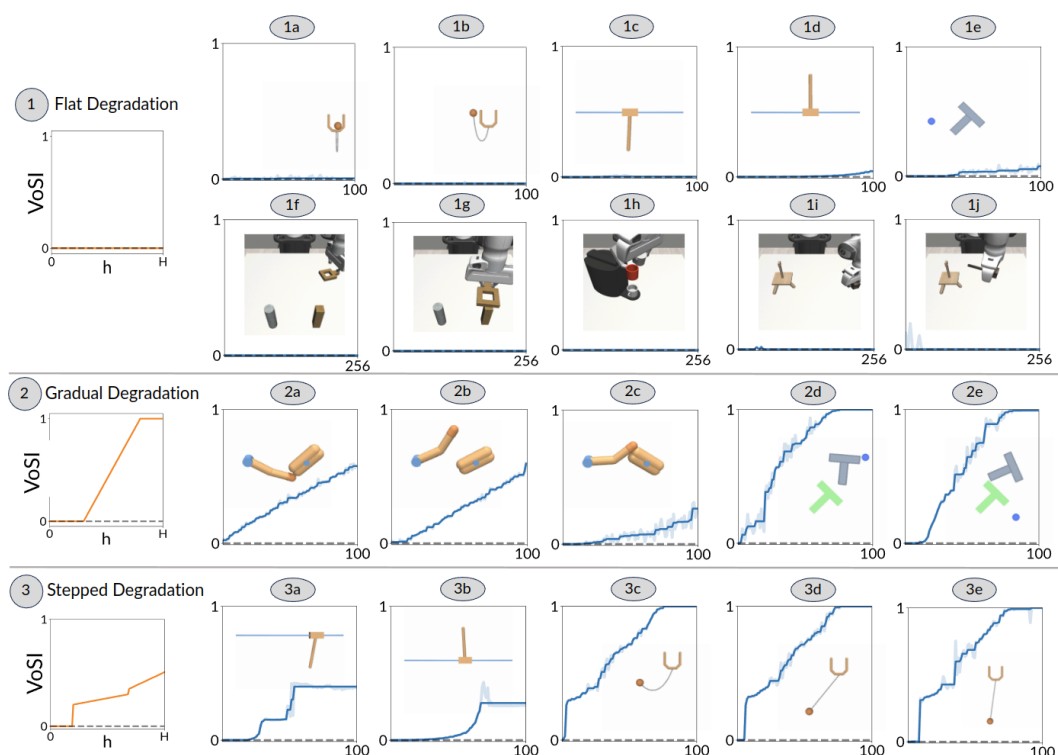

Figure 7: **VoSI profiles** of states encountered in our experiments on 7 robotic tasks can be broadly sorted into three prototypical profile shapes, shown here as 1) flat, 2) gradual, and 3) stepped profiles.

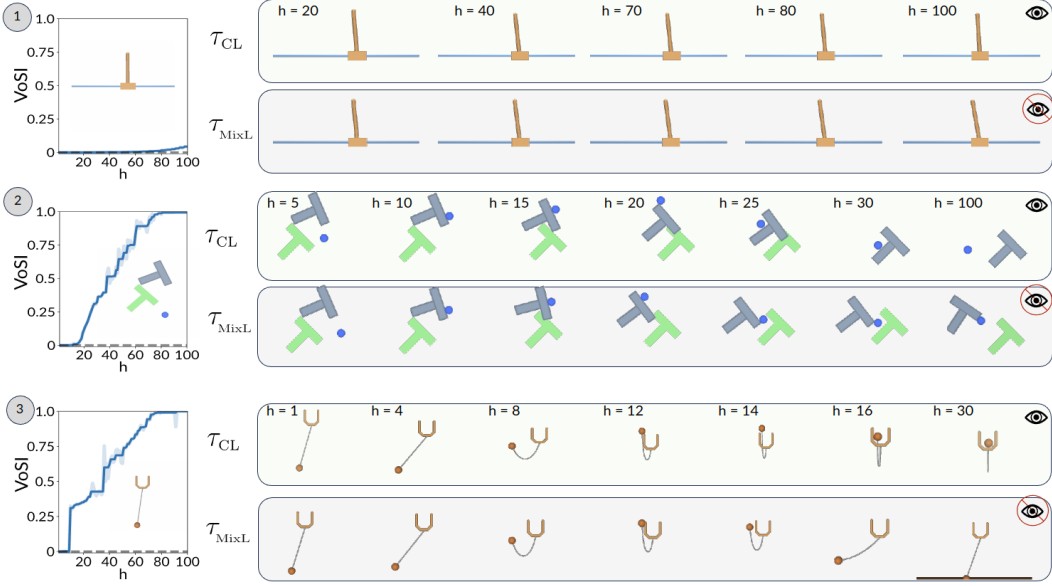

Figure 8: Time-aligned visualization of one closed loop trajectory $\tau_{\mathrm{CL}}(s_t)$ and the open loop phase of one mixed loop trajectory $\tau_{\mathrm{MixL}}(s_t, h)$, starting from an initial state $s_t$. The VoSI profile of $s_t$ is plotted on the left, and indicates a measurement of the task-relevant deviation between $\tau_{\mathrm{CL}}$ and $\tau_{\mathrm{MixL}}$.

▶ **"Stepped"** VoSI profiles (Figure 7-3) involve sharp increases at some values of $h$ followed by steady increases or flat stretches. These are indicative of phase changes in the task. For example, in cup-catch(Figure 7-3c-e), failing to closely track the ball's complex dynamics as it is dynamically

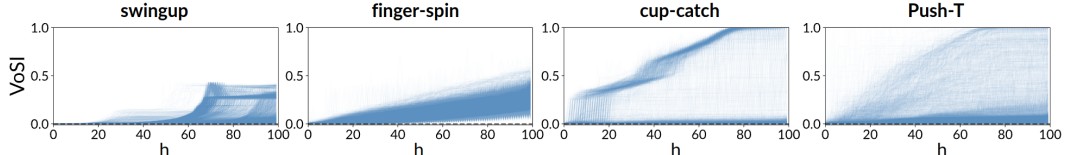

Figure 9: Visualization of all the VoSI profiles overlayed for four tasks.

thrown up can can make the difference between the ball landing in the cup, or instead colliding with its lip. Similarly, in `swingup`, not sensing during a short and dynamic phase can prevent the agent from correcting for overshoot or undershoot, causing sharp reward losses.

Finally, in Figure 9 we take a step back and overlay the VoSI profiles for all states encountered in $\tau_{\text{CL}}$ executions on each task. These plots reveal information about characteristic VoSI profiles in each task. Further, the dark "gradual" and "stepped" curves in `finger-spin` and `swingup` correspond to frequently repeated states in these tasks involving periodic movements.

**Observing VoSI Evolution Over Time.** Our analysis above has focused on individual states sampled from across all tasks, but we now plot the evolution of VoSI profiles of states encountered in a closed loop rollout of the `cup-catch` task in Figure 10. Each column here corresponds to the VoSI for a specific time-step in the following manner: the x-axis corresponds to the time $t_1$ at which the VoSI profile $\text{VoSI}(s_{t_1}, h)$ starts, and the y-axis represents absolute times $t_2 = t_1 + h$ at which the VoSI is evaluated. In other words, $(t_1, t_2)$ in the plot corresponds to $\text{VoSI}(s_{t_1}, t_2 - t_1)$. Each row of this visualization thus lines up the VoSI measurements for a future time instant $t'$, from all past time instants $t < t'$.

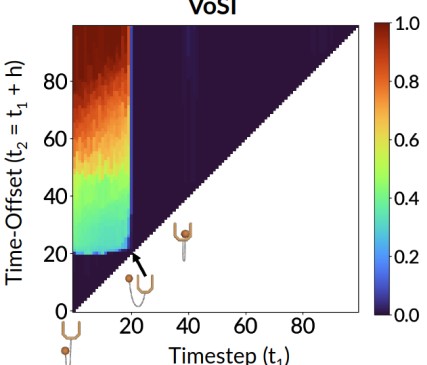

Figure 10: VoSI profiles over a trajectory.

This makes it possible to easily evaluate temporal consistency trends. For example, all states $t_1$ prior to $t = 20$ largely "agreed" on the same VoSI values to assign all states. This consistency indicates that no task-relevant information was sensed during $0 < t < 20$. Further, the large VoSI values assigned to futures beyond $t = 20$ at all times before that moment indicate a kind of event horizon. Sure enough, VoSI estimates at timesteps after $t = 20$ diverge from those before: the moment the agent had sensed at $t = 20$, its value for sensing at all future moments dropped to 0. Rare critical information was observed at $t = 20$. We overlay visualizations of the states at $t = 0, 20, 40$ over the plot: $t = 20$ represents a tipping point after which the ball is destined to fall into the cup, so that no further sensing is necessary. We provide more details and examples of such plots in Appendix E.

**Reward-Free and Interaction-Free Measures of VoSI.** As introduced in Equation (1), VoSI requires task rewards, and must be computed through additional interaction with the environment starting from each state. To make our analysis accessible outside of simulated RL benchmark tasks where these conditions can be easily met, we study two alternative implementations of VoSI, explained in detail in Appendix C. Briefly, the *state disagreement VoSI-S* measures the mean state-wise distance of mixed loop trajectories from closed loop trajectories. This requires no reward function, but still requires the execution of the mixed-loop trajectory. When no more interaction with the environment is possible, we propose the *plan disagreement VoSI-P*, which measures how much the look-ahead policy $\pi$'s action plans formed at $s_t$ differ from actions executed in the closed loop.

Table 1 shows that these alternatives generally have high rank correlation with the reward loss-based VoSI from Equation (1) (VoSI-R). As we show in Appendix C.1, disagreements can often be traced down to some degen-

| correlation | swingup | cup-catch | finger-spin | push-T |
|---|---|---|---|---|
| $\rho(\text{VoSI-R}, \text{VoSI-S})$ | 0.78 | 0.88* | 0.98 | 0.53* |
| $\rho(\text{VoSI-R}, \text{VoSI-P})$ | 0.54 | 0.18* | 0.73 | 0.37* |

Table 1: Correlations between VoSI metrics.

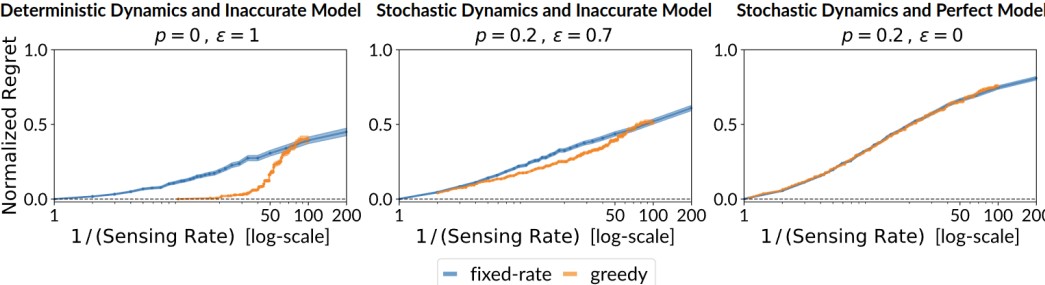

Figure 11: Visualization of task regrets profiles achieved for different scenarios on `four-rooms` by the fixed-rate mixed loop execution strategy and greedy strategy utilizing the state-wise VoSI profiles.

erate phases of a task, such as after the T is already at the target position in `Push-T` so we discount them from computing the correlation[*]. At this time, agent actions do not matter to task rewards (as measured in VoSI-R), but can generate potentially large action disagreements (VoSI-P).

**Towards Synthesizing Efficient Execution Strategies.** Our primary goals in this manuscript have been to develop a new toolkit to study foundational scientific questions in robotics and decision making, and get the first answers in some settings of practical interest. Answers to these questions have many potential practical uses, and we now showcase a simple implementation that is within reach given the extensive analyses above: synthesizing execution strategies for a look-ahead policy $\pi$ that are efficient efficient in terms of sensing and other entailed resource costs.

We present a simple greedy strategy that sets the open loop execution length $h = \eta_{\text{greedy}}(s; (B, T))$, where $s$ is the currently observed state, $B$ is the task regret budget (how much reward we are willing to lose relative to closed loop), $T$ is the remaining episode length. Starting after sensing the state $s_0$ at time $t = 0$, $\eta_{\text{greedy}}$ sets the largest horizon $h$ such that $\text{VoSI}(s, h) < \frac{hB}{T}$. It then decrements the budget $B$ and time $T$ by $\text{VoSI}(s, h)$ and $h$ respectively (see Algorithm 1 for more details). Varying the initial budget available to $\eta_{\text{greedy}}$ generates a family of execution profiles with varying degrees of parsimony (measured as the average sensing rate achieved). For evaluation, recall the task regret analysis in Figures 3 to 5, where we plotted the normalized regret vs. sensing rate of fixed rate executions with varying fixed $h$. In Figure 11, we compare the $\eta_{\text{greedy}}$ family against fixed rate executions on variations of the `four-rooms` task from Section 4.1. $\eta_{\text{greedy}}$ effectively exploits VoSI to achieve better performance at most sensing rates than the naive fixed rate strategy. We compare the proposed strategy with a baseline that is designed to reduce sensing requirements in Appendix H.

## 6 DISCUSSION AND CONCLUSIONS

We have presented a framework to empirically investigate sensory requirements of sequential decision-making agents through a value of information-inspired lens. Our framework permits a first investigation of what sensory information matters and how much, to state of the art policy architectures, in representative robotic tasks. The insights here offer a first glimpse of the value of sensory information, and the efficiencies that understanding it can enable.

However, our empirical study is still only a small step towards understanding the sensory needs of robots. The VoSI framework in this paper is limited to a particular kind of robot: one that is provided with a pre-trained closed loop look-ahead policy $\pi$, and is then charged with making decisions about when to sense such that the policy performance is not overly affected by the omitted sensory inputs. As such, while we might like to have insights on the value of sensory information based on the *best possible behavior* under each sensing pattern, our findings can only approximate this through what $\pi$ can achieve. Even beyond this, we have studied *when* robots need to sense their environments under the most generous assumptions on the task and sensing apparatus: largely deterministic environments except in a toy gridworld setting, and noise-free instantaneous actuation and sensing. In Appendices, we have included proof-of-concept examples of VoSI analyses: in a cartpole swingup task with noisy vision-based state estimation (Appendix I), and in a multi-sensory realistic quadruped climbing task (Appendix J). Future work could further widen the scope of real-world tasks for studying VoSI, and study how it can inform efficient policies.

ACKNOWLEDGMENT

This research was supported in part by NSF CAREER Award 2239301, NSF Award 2331783, DARPA TIAMAT (HR00112490421), and a generous gift from AWS AI to the ASSET Center for Trustworthy AI at Penn Engineering. The authors would like to thank the members of the Perception, Action, & Learning group at Penn for the insightful discussions, and the ICLR reviewers for their valuable and constructive feedback.

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

TABLE OF CONTENTS

## A  SETUP AND HYPERPARAMETERS

We additionally encourage the reader to visit the project website at `https://sites.google.com/view/vosi-robotics/` to view more task-specific details and explore visualizations that highlight different slices of the data beyond the ones presented in the main paper. The code used to conduct the experiments will be made available on the website.

### A.1  TD-MPC2

We use a JAX implementation of the TD-MPC2 algorithm and retain the default hyperparameters (listed in Table 2) associated with training the TD-MPC2 – except for the experiments with varying model-sizes for which we alter the hidden dimensions of the neural architecture. This reproduction closes matches and exceeds the performance of the original PyTorch implementation. We obtain performant TD-MPC2 checkpoints for the dense-reward DM-control suite tasks considered in 200K steps and the Push-T task was trained up to 2M steps with an additional modification of clipping the output actions to not move than 50 pixels in the PyMunk interface of the Push-T task to generate smooth motion profiles as the performant RL trajectories were used as seed data for training the Diffusion Policy representations on these tasks. TD-MPC2 failed to produce meaningful behavior on the robosuite tasks potentially due the sparse reward nature of the task. To execute the TD-MPC2 agent in mixed-open loop manner we disable warm-starting the model-based search and just rely on the latest state observation and use the policy prior to seed the action plan search.

#### A.1.1  CHECKPOINT SELECTION FOR MODEL-STAGE AND MODEL-SIZE EVALUATION

To test the hypothesis of how model finiteness and expressivity influence the sensing needs of a TD-MPC2 agent. We design two studies on the Push-T and finger-spin tasks: (1) select checkpoints at different stages of training (Table 3) (2) synthesize expert controllers starting from different representational capacities (keeping all other hyperparameters the same) (Table 4).

Table 2: Default hyperparameters of TD-MPC2 agent

| Parameter | Value |
|---|---|
| encoder.layers | [256, 256] |
| encoder.lr | 1e-4 |
| world_model.mlp-dim | 512 |
| world_model.latent-dim | 512 |
| world_model.value-dropout | 0.01 |
| world_model.num-value-nets | 5 |
| world_model.num-bins | 101 |
| world_model.symlog-min | -10 |
| world_model.symlog-max | 10 |
| world_model.lr | 3e-4 |
| world_model.max-grad-norm | 20 |
| mpc.horizon | 3 |
| mppi-iterations | 6 |
| population-size | 512 |
| num-elites | 64 |
| min-plan-std | 0.05 |
| max-plan-std | 2 |
| temperature | 0.5 |
| optim.warmstart | false |
| optim.batch-size | 256 |
| optim.discount | 0.99 |
| optim.rho | 0.5 |
| optim.consistency-coef | 20 |
| optim.reward-coef | 0.1 |
| optim.continue-coef | 1.0 |
| optim.value-coef | 0.1 |
| optim.entropy-coef | 1e-4 |
| optim.tau | 0.01 |

Table 3: TD-MPC2 checkpoints selected for characterizing task-regret profiles at different stages of learning

| Task | Level | Steps | Return |
|---|---|---|---|
| finger-spin | beginner | 14K | 44.37 |
| | novice | 100K | 117.39 |
| | expert | 200K | 133.25 |
| Push-T | beginner | 200K | 25.62 |
| | novice | 1M | 58.53 |
| | expert | 2M | 66.59 |

## A.2 DIFFUION-POLICY (DP)

We use the code adopted from the official repository (Chi et al., 2023) for training U-Net based DP agents and retain most of the hyperparameters and configurations from the original repository to train policies on the state-based tasks (Table 5) – the only key difference is the extended prediction horizon lengths (close to the episode length) used for training, this was done in order to synthesize mixed loop executions with increasingly lower sensing rates. The Robosuite tasks are obtained from MimicGen repository (Mandlekar et al., 2023) as it came with over 200 high quality demos for each task square-nut, coffee-pod, and threading – all requiring some degree of precise insertion. The success rates over 100 rollouts on these tasks are reported in Table 6. On the DM-Control suite and Push-T task we train Diffusion Policy agents on the dataset of 200 trajectories from

Table 4: TD-MPC2 checkpoints selected for characterizing task-regret profiles at different model capacities

| Task (Steps Trained) | Level | enc.mlp-dim | wm.mlp-dim | wm.latent-dim | Return |
|---|---|---|---|---|---|
| finger-spin (200K) | small | 64 | 64 | 32 | 124.33 |
| | medium | 128 | 256 | 128 | 136.74 |
| | large | 512 | 1024 | 1024 | 137.92 |
| Push-T (2M) | small | 64 | 64 | 32 | 33.833 |
| | medium | 128 | 256 | 128 | 70.063 |
| | large | 512 | 1024 | 1024 | 73.704 |

closed loop execution of the performant TD-MPC2 policy and were able to synthesize meaningfully performant policies on swingup and Push-T but failed on finger-spin and cup-catch tasks. We suspect that the non-smooth nature of the control signals of RL-controllers used to supervise the imitation learning agent as potential reason for this, however note we did not perform an extensive hyperparameter search to try to synthesize the best policies. The issue of non-smooth target control signals for imitation is ameliorated in swingup with an explicit control cost penalty when training expert TD-MPC2 agents and on Push-T task with the actions constrained to generate smooth motions beneficial for imitation.

Table 5: Default hyperparameters of Diffusion Policy for a task of episode length $T$

| Parameter | Value |
|---|---|
| input_embed_dim | 256 |
| step_embed_dim | 256 |
| encoder_layers | (512, 512) |
| unet.down_dims | (256, 512, 1024) |
| kernel_size | 5 |
| n_groups | 8 |
| num_diffusion_steps | 100 |
| ema_power | 0.75 |
| dim | 256 |
| num_demos | 200 |
| train_epochs | 3000 |
| batch_size | 256 |
| lr | 1e-4 |
| weight_decay | 1e-5 |
| pred_horizon | $\min(300, T)$ |
| obs_horizon | 1 |
| act_horizon | 4 |

Table 6: Success Rates for DP agents on Robosuite tasks

| Task | Success Rate (%) |
|---|---|
| square-nut | 79 |
| coffee-pod | 86 |
| threading | 80 |

### A.3 TASK-REGRET PROFILE AND VOSI PROFILE EVALUATION PROTOCOL

We evaluate fixed-rate mixed loop execution strategy for both the lookahead policies on all values of the fixed-rate execution horizons $h = \{1, \cdots, T\}$ where $T$ represents the episode length of the task and report the average return obtained over 100 rollouts of the mixed loop execution.

**Implementing VoSI Profiles.** For both TD-MPC2 agent on DM-control and Push-T tasks, we start by collecting a buffer of states from 50 rollouts of closed loop executions of the policy. For each state $s$ in this buffer we seek to characterize $\text{VoSI}(s, h)$, we do this by reseting the simulator to that state and then run mixed loop strategies $\eta_{\text{MixL}}(s, h)$ with the open loop length $h$ varying from 1 to 100 steps. For each $h$, we generate 5 full trajectories (i.e., length of trajectory = episode length $T$) from $s$. Now this permits a Monte Carlo estimate of the expectations over $\tau_{\text{MixL}}(s, h)$ in the second term in Equation (1) – the variance of the values are not too high as we are starting from a performant underlying policy. For the expectation over $\tau_{\text{CL}}$ we similarly sample 5 trajectories from closed loop execution of $\pi$ and estimate this quantity. Thereby yielding an estimate of $\text{VoSI}(s, h)$. This is a computationally expensive procedure especially for evaluating long-horizon policies as for each state we generate on the order of 500 trajectories, which starts becoming prohibitively expensive to run a DP agent for given the higher inference costs. We therefore, examine the closed loop trajectories from the lens of the plan-disagreement metric (Appendix C) to identify states where the policy reacts to sensory information (which happens only at regions of object interaction as observed in Figure 12) and perform the counterfactual examination only on states preceding such states alone – effectively filtering out states where there is extended free-space motion of the manipulator.

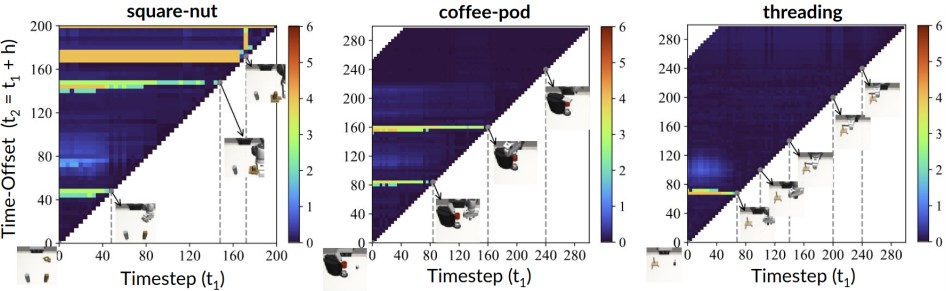

Figure 12: Visualization of the planning disagreement metric (VoSI-P) aligned along timesteps of representative rollouts of the Diffusion Policy agent on robosuite tasks. These identify characteristic object-interaction points near which expensive counterfactual return-based metrics are evaluated.

## B  GRID WORLD

**Setup**  As identified in Section 4.1 there are two scenarios that justify processing environment feedback: (1) stochastic dynamics (2) inaccurate model of the dynamics. We perform a set of controlled experiments on the `four-rooms` grid world and study the sensing needs of a model-based controller that has a deterministic optimal closed loop policy $\pi^* : S \rightarrow A$ identified for each environment with stochasticity ($p$) by value-iteration and a dynamics world model $\hat{P}$ that is used to sample lookahead action sequences. To study the axis of model suboptimality we simulate a representation error $\epsilon$ in the dynamics model of the agent by finding transition probability matrix $\hat{P}$ s.t. $D_{\text{KL}}(P(s'|s,a), \hat{P}(s'|s,a)) = \epsilon \ \forall \ s,a$. We identify such $\hat{P}$ by simply setting $\hat{P}(s'|s,a) = c(s,a) \cdot U(s'|s) + (1 - c(s,a)) \cdot P(s'|s,a)$, where $U(s'|s) = \frac{\mathbb{1}_{[s' \in N(s)]}}{|N(s)|}$ denotes a uniform prior over the reachable neighbors $N(s)$ (as an effect of some action) of state $s$. The weighting term $c(s,a) \in [0,1]$ is identified to satisfy the constraint on KL-divergence.

**Task-Regret Profile and VoSI Profile Evaluation Protocol**  The task-regret profiles (Figures 4 and 11) are obtained by evaluating the performance of the execution strategies over 1000 rollouts upto a horizon of 200.

The VoSI profiles used by the greedy execution strategy (Algorithm 1) are monte-carlo estimates of the expected reward loss (Equation (1)) for each state $s$ in the grid world by simulating 1000 rollouts of $\tau_{\text{MixL}}(s,h) \ \forall h \in \{0, \cdots, 100\}$. Observe that $\tau_{\text{MixL}}(s,0)$ denotes the distribution of closed loop execution trajectories $\tau_{\text{CL}}$. Therefore once rollouts of $\tau_{\text{MixL}}(s,h)$ are independently sampled the value-of-sensing information $\text{VoSI}(s,h)$ in a $h$ time-window after observing $s$ can be estimated.

---

**Algorithm 1** Greedy mixed loop execution Strategy ($\eta_{\text{greedy}}$)

---

**Require:**
  $\pi$      : a policy that can produce lookahead actions
  $B$      : a budget
  $\text{VoSI}(s,h)$ : the value of sensory information profiles
  $T$      : the episode length

  $\triangleright$ Strategy is to use the budget the equally distribute the cost of acquiring sensory information over the episode and commit to a greedy strategy

  $t = 0$
  **while** $t < T$ **do**
    $\delta = \frac{B}{T-t}$
    $h \leftarrow \max h \ \text{s.t} \ \text{VoSI}(s_t, h) \leq h \cdot \delta$
    $B \leftarrow B - \text{VoSI}(s_t, h)$
    $t \leftarrow t + h + 1$
    Execute($\pi_{0:h}(s_t)$)
  **end while**

---

## C  DETAILS OF THE REWARD-FREE AND INTERACTION-FREE MEASURES OF VOSI

To expand the scope of VoSI measures to leverage trajectory discrepancy characteristics beyond just task rewards. We study a few simple alternate measures:

- *State Disagreement* $\text{VoSI-S}(s,h) := \mathbb{E}_{\tau_{\text{MixL}}(s,h)} \left[ \min_{\tau_{\text{CL}}(s)} \frac{1}{T} \sum_{t=0}^{T} ||s_t^{\text{CL}} - s_t^{\text{MixL}}||_2 \right]$ measures a simple time-aligned state discrepancy of measure of distribution $\tau_{\text{MixL}}$ from $\tau_{\text{CL}}$. One can more broadly use any optimal-transport based formulation to characterize these state distribution disagreements as used in Luo et al. (2023); Haldar et al. (2023) to obtain a reward signal from a distribution matching perspective.

- *Plan Disagreement* $\text{VoSI-P}(s, h) := \mathbb{E}_{\tau_{\text{CL}}(s), \hat{a}_{0:h} \sim \pi_{0:h}(s)} \left[ \frac{1}{h} \sum_{t=0}^{h} ||a_t^{\text{CL}} - \hat{a}_t||_2 \right]$ a measure akin to the imitation-loss which suggests the deviation of action plans from closed loop actions sequences executed in the environment. Such a metric does not need access to the distribution of $\tau_{\text{MixL}}$ trajectories and can therefore be applied on a dataset of $\tau_{\text{CL}}$ trajectories.

### C.1 ON THE ERRONEOUS CORRELATION OF METRICS DUE TO TASK-COMPLETION

By visualizing state-aligned orderings of different alternative VoSI metrics in Figures 13 and 14, where each row across the subplots correspond to the same state $s$ with a few representative states visualized in the buckets. We observe that on states that denote task-completion or states near task completion, have a close to flat VoSI-R profile, but exhibit varying degrees of VoSI-P or VoSI-S due to stochasticity of the policy post task-completetion which bears no consequence on the reward achieved but can make the estimates of other trajectory disagreement metrics relying on states (VoSI-S) and actions (VoSI-P) quite noisy and thereby does not provide much signal. We therefore discount such states when presenting the rank-order correlation statistics in Table 1 to suggest how effective alternative metrics can be in computing notions similar to VoSI-R.

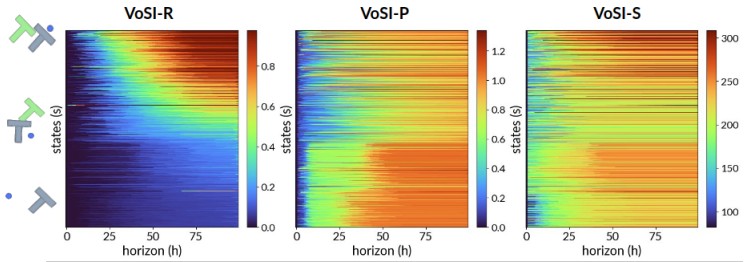

Figure 13: Push-T

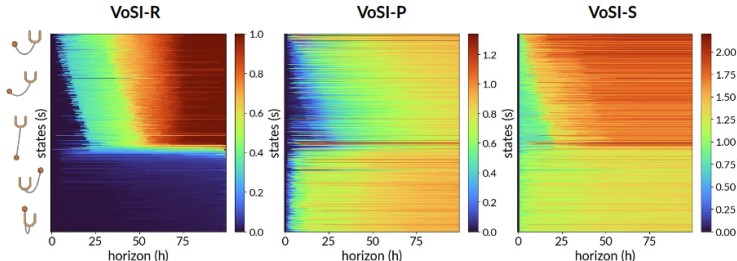

Figure 14: cup-catch

## D COMPARING DIFFERENT POLICY REPRESENTATIONS

**The Impact of Policy Architectures.** From Figure 3 in Section 4 we observed for tasks `swingup` and `Push-T` for which we have both TD-MPC2 and DP policies, that the planning based TD-MPC2 agent exhibited lesser degradation in task performance when constrained to operate at much lower sensing rates in comparison to a reactive DP policy. To gain a finer grained understanding of the sensory dependence of these policy representations we employ our method of analysis and compute state-wise regret profiles of these different policy representations on common task states and illustrate characteristic differences in Figure 15. In `swingup`, we observe that DP exhibits much higher performance degradation in the swingup phase of the task when compared to TD-MPC2, but degrading lesser during the balance phase of the task. For `Push-T`, we observe that DP policy critically relies on sensing to retain some performance in comparison on TD-MPC2.

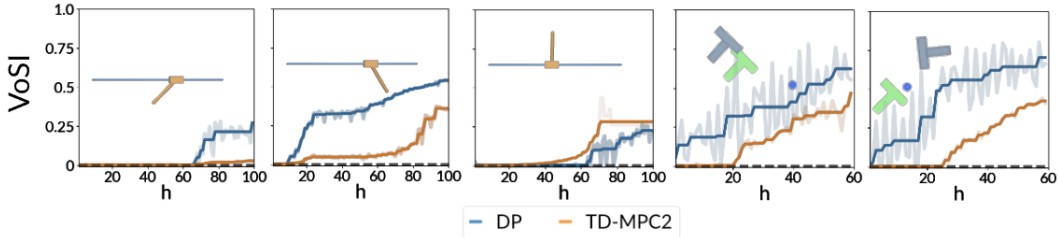

Figure 15: Visualization of VoSI profiles of TD-MPC2 (planning-based) and DP (reactive-controller) along a few common task states.

# E DISCUSSION ON VOSI PROFILES OVER A TRAJECTORY

In Figure 9, we visualize overlayed VoSI profiles (with transparency alpha = 0.02) for states sampled uniformly from closed loop rollouts of a fixed episode length. In tasks like cup-catch and Push-T the task objective is achieved much before the episode ends i.e. the ball is in the cup or the T object is aligned with the goal. At this point, VoSI profiles are flat because actions are trivial and the policy does not perform further meaningful actions. This is why there is a high density of seemingly flat profiles for these tasks in Figure 9.

For most figures in the paper we have visualized the evolution of VoSI profiles over the open loop horizon $h$. In Figure 10, we additionally convey how the VoSI profiles $\text{VoSI}(s, h)$ evolve not only over open loop horizon $h$ but also over the states $s$ (starting from $s_0$ to $s_T$ of a closed loop rollout). This means we are mapping the scalar $\text{VoSI}(s_{t_1}, h)$ of two arguments (time index $t_1$ of last observed state $s_{t-1}$, and open loop horizon $h$). We use colors to indicate the VoSI values. In Figure 10, the x-axis represents the time-index $t_1$ of the last observed state $s_{t_1}$ and y-axis represents the offset open-loop horizon $t_2 = t_1 + h$. Each vertical column of colors represent VoSI profiles for a particular state at varied horizons. The choice of adding the last-observed time offset in the y-axis i.e. using $t_1 + h$ rather than $h$ helps highlight the structure in the evolution of VoSI over time. Tracking along the horizontal row $t_2$ we can interpret how valuable re-sensing before $t_2$ is at prior moments in time.

We provide more illustrative visualizations of the evolution of VoSI profiles over the course of a rollout akin to Figure 10 in Figure 16 for the swingup and finger-spin tasks. We observe characteristic periodicity in VoSI profiles along the trajectory in the balance-phase of the swingup task and peak-spinning phase of finger-spin i.e. any state in that window has similar VoSI profile indicating that one could execute fixed-rate mixed loop executors in such windows to sense at the boundaries of where there is value to gain from sensing and repeat.

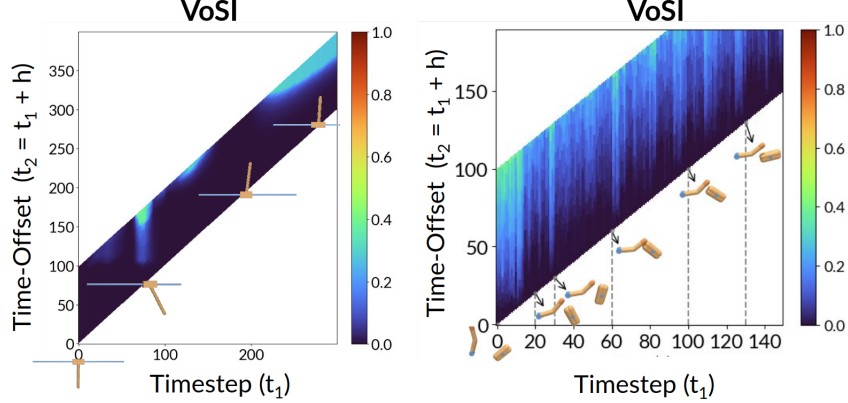

Figure 16: Evolution of VoSI profiles over a trajectory for swingup **(left)** and finger-spin **(right)**

# F  IMPACT OF TRAINING STAGE AND MODEL CAPACITY ON SENSING NEEDS OF THE POLICY

Figure 5 shows a drop in performance of mixed-loop execution relative to closed-loop performance for each policy, but closed-loop performance for beginner and novice are often poor, meaning that there is "less to lose" when executing mixed-loop. For example, in the Push-T task, our "beginner" policy executed closed-loop gets a return of 25.62 compared to 66.59 for the expert. Thus, even if "beginner" deteriorates to a lower return with mixed-loop execution, this registers in the "relative regret" performance metric as a relatively small drop in task regret.

In Figure 17, we provide a version of the plot where the regret scales are the same by computing the task performance regret with respect to the performance of the "expert" policy checkpoint for training stage (and "large" checkpoint for the model capacity experiment). From Figure 17, it becomes more evident that the expert is capable of retaining higher levels of performance as sensing rate decreases.

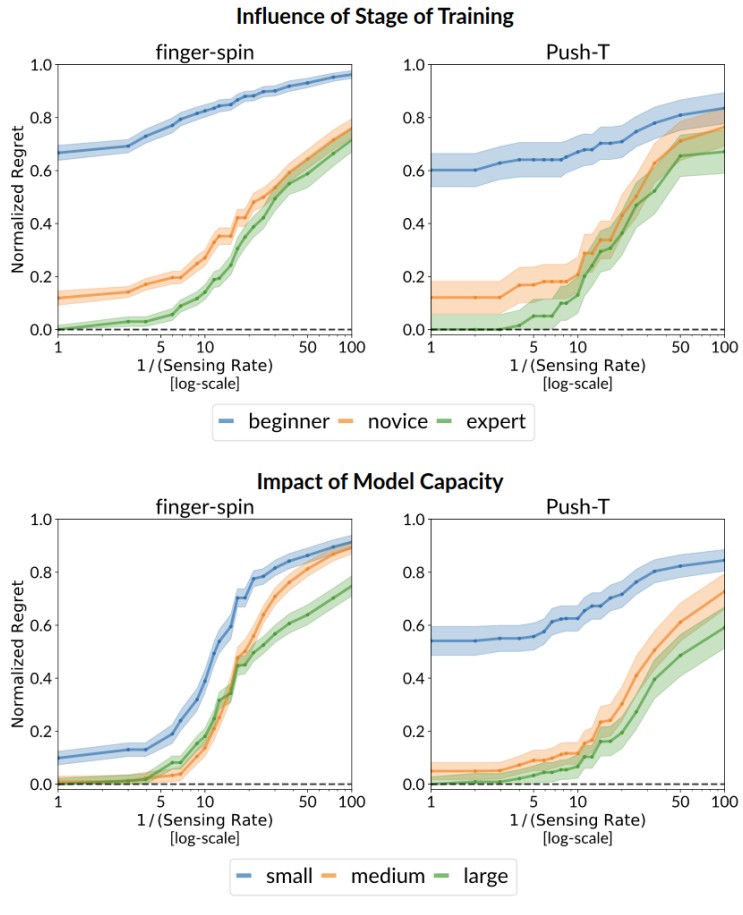

Figure 17: Task regret profiles on two tasks with respect to the 'expert' (**above**) and 'large' (**below**) policy checkpoints ensuring that scales of the regret profiles are commensurate.

## G    EXPANDED RELATED WORK

The questions studied in this work are a subset of the questions that Donald (1995) first posed to the robotics community, asking what information is needed for a robot to perform a task, how the robot might acquire such information, and more. Their initial steps towards answering these questions developed the notion of "information invariants", which describe loosely speaking, level sets of task performance in terms of sensed or stored information and other resources such as speed and communication bandwidth among multiple agents. These foundational questions still persist Koditschek (2021). One prominent line of inquiry is on sensor reductions or equivalent classes and sensor dominanceLaValle (2019), which describes a framework for computing partial orderings amongst sensing setups for robots in a task-agnostic manner. Related, Zhang & Shell (2020) provide a framework for searching jointly over sensor designs and plans for some task, and is demonstrated on simple tasks with a small number of discrete states and actions. Erdmann (1995) set up specialized minimalistic sensors for a policy that "sense not the states but the applicable actions", they do so by starting from abstract idealized sensor to identify all states where a given action is guaranteed to make some progress in a task (as tracked by a change in progress-measure or value function defined on the state space). The applicable action-based subsets form a cover of the state space. The author then proposes task-specific minimalistic "action-based" sensor that is designed to capture just enough information of the state to identify atleast one action that is guaranteed to make progress in a task, essentially setting up a minimum cover problem. Note that in the process of the sensor design we have consequently also obtained an applicable action for each state – with the guarantee that the strategy accomplishes the task under the sensor designed. More recently, McFassel & Shell (2023) have expanded the applicability of these ideas by increasing flexibility in the way progress measures are defined. The foundational framework laid out by Erdmann (1995) serves as a powerful tool that provides insight into minimalistic sensor design, it is limited to systems that can be fully modeled and analyzed analytically under restricted policy classes. While the framework provides insights into what to sense at every timestep for a closed loop policy and does not characterize when acquiring this information is critical. Furthermore, it does not characterize the degradation in performance achievable under a certain sensor design.

In control theory, the questions of devising control systems that are parsimonious with processing the full state information are studied extensively by prior works that propose event-based sampling and event-triggered control (Åström & Bernhardsson, 1999; Astrōm, 2008; Vasyutynskyy & Kabitzsch, 2010; Heemels et al., 2012). Such systems are designed to change the prescribed control plan only when *events* (discrete variables that are a function of a partial state of the system) are triggered. The event-triggered control literature, much like the works discussed in Section 2, aim to generate controllers that operate with limited sensing. Event-triggered control typically (1) assumes that a low-cost sensor is constantly monitoring the state at each time $t$ and (2) defines a "trigger" event when a manually defined function of this sensory observation crosses a threshold. For example, in Trimpe & D'Andrea (2011) an event trigger is defined based on the deviation of the coarsely sensed state from a model's predictions. In contrast, in this work we have sought to characterize for an arbitrary lookahead policy a notion of its reliance on external feedback in-terms of its task performance. The insights from which can complement the event-triggered control literature by aiding with the design of events and triggers for efficient execution of the policies.

Outside of robotics, in machine learning and information theory, Tishby & Polani (2010), like us, present an information theoretical view of sequential decision making: they define the "information-to-go" property of a policy $\pi$ based on the KL divergence from a prior, of the distribution of futures generated by $\pi$. They show that this quantity follows a Bellman-like recursion, which is useful to optimize "informationally constrained" policies that maximize task rewards while generating constrained information-to-go. These constraints are not directly on any information processing capabilities on the agent, but rather on "the information processed in the joint agent-environment system", i.e., the surprisingness of policy-induced trajectories relative to the prior. The priors considered are uninformative, and their information-to-go vanishes when the dynamics is fully known. Our VoSI is affected by stochasticity even in fully known dynamics, as well as the agent's limited information processing capabilities (see Section 4.1 for a discussion). Finally, they don't deal with when / how the information should be acquired, which is central to our approach. Their work's influence appears in recent works Eysenbach et al. (2021); Lu et al. (2023) that apply similar constrained optimization techniques to generate policies that must extract less of the sensed information, but without any constraints on when sensing is available. Closer to us in this family of

work is Van Dijk et al. (2009), who demonstrate through evolutionary optimization that temporally extended options-style architectures are preferred when information processing is constrained. Fox & Tishby (2015) extends Tishby & Polani (2010)'s framework to POMDPs: as their notion of information-to-go, they measure deviation from a prior that includes an ergodic action distribution that is loosely related to the open-loop policy executions we consider in this work. Lu et al. (2023) motivate the study of what information an agent should sense and retain for maximizing data efficiency during online reinforcement learning, where the data is ever-changing. They motivate an analysis of the cost-per-bit of information (measured in entropy) sensed to benefit obtained (task return), and conduct experiments on online learning in bandit settings to demonstrate implications of agent designs that factor in the cost of acquiring and retaining information for learning polices that minimize the regret accrued in trying to reach a stationary target policy.

Finally, decision making is also of interest in economics, and sure enough, our questions have precedents that are studied in this discipline Maćkowiak et al. (2023), starting from the foundational work of Sims (2003). "Rational inattention" posits that (human) agents cannot process all available information but can choose which pieces of information to attend to, explaining many observed macroeconomic phenomena.

## H A POINT OF COMPARISON FOR THE REDUCED SENSING APPLICATION

Figure 11 shows a proof-of-concept for a potential application of VoSI to reduce sensing needs when executing a pre-trained policy. We use a simple greedy approach that can adapt to test-time-specified sensing costs. To offer a point of comparison for the VoSI-based greedy efficient execution strategy (described in Section 5) we consider an "event-triggered control" based approach described below:

**Event-triggered approach**  We note that "event-triggered control" literature (see Appendix G) is relevant here as an approach that reduces the amount of expensive state sensing. However, note that it does not completely forgo sensing at any moment but instead chooses to sense at a coarser fidelity until "an event is triggered". For the `four-rooms` task, we define coarse sensors that identify the agents position in different sub-grids depicted in Figure 18 **(left)**. We "trigger" an event to sense the true MDP state when (a) the coarse sensor recognizes a transition between sub-grids, or (b) no change is detected by the coarse sensor within a specified timeout (a hyperparameter) when it is not within the goal region (green sub-grid). As the timeout parameter is varied for this baseline, we obtain the regret profile by measuring the task performance vs average sensing rate (which only includes the number of times the high fidelity state observation is queried).

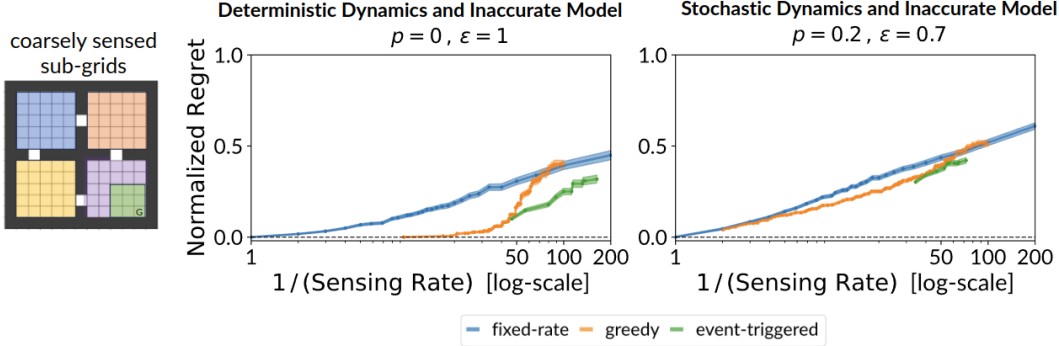

Figure 18: Task regret profiles achieved for `four-rooms` by different execution strategies. Note that the event-triggered approach here coarsely senses the state of the environment at every step i.e. the agent's presence in some sub-grid depicted on the **left**. The proposed greedy strategy or the fixed rate execution baseline does not have access to such additional information.

From Figure 18, we observe that the event-triggered approach achieves higher task performance (lower regret) compared to our proposed greedy strategy when constrained to operate at lower sensing rates. The extra knowledge, in the form of coarse sensing, proves to be a significant advantage for the event-triggered approach in the deterministic dynamics setting (Figure 18-middle), but when dynamics is stochastic, our naive VoSI-based greedy strategy matches this approach in performance, even without the sub-grid sensor. The coarsely sensed "room" based event triggers might not be very beneficial in environments with stochastic dynamics as the action plans necessary within a sub-grid might change widely as a result of noise in transition dynamics – this results in the event-triggered approach relying more on the heuristic timeout to obtain high-fidelity state observations.

# I  CASE STUDY: NOISY SENSING

We would like to demonstrate the applicability of the proposed VoSI analysis in settings beyond the scenarios primarily studied in the paper i.e. deterministic environments with perfect state sensing. We provide a proof-of-concept application of VoSI as a probing tool to study an agent that operates under noisy sensing on the `swingup` task.

We train a small CNN-based visual state estimator that has up to 5 cm and 2 degrees errors in the predicted cart position and pole angle, which is quite significant in this task. We then obtain a TD-MPC2 policy that operates on these state estimates (see Appendix I.1 for more details) and conduct a similar VoSI analysis (Section 5). As such, the value of new sensory measurements increases: this is reflected in steeper VoSI profiles overall. However, trends remain similar: for example, we see stepped VoSI profiles during the swingup phase, and close to flat VoSI profiles even up to 70 steps of open-loop execution during the balancing phase when the pole is already close to vertical. See Appendix I.2 for more details.

This establishes both that our VoSI framework is applicable to many more general task setups than reported in the main paper, and that the findings reported in the main paper already offer useful intuitions for what results in more general/complex task setups might look like.

## I.1  TRAINING THE STATE ESTIMATOR AND POLICY

To create a dataset for the state estimator, we sample 20000 configurations with cart position (in meters) $\sim U(-2, 2)$, pole angle (degree) $\sim U(-180, 180)$. At each of these configurations we capture a (128, 128, 3) RGB image from a front-view camera positioned 6.5m from the center of the rail. Figure 19 shows some sample front view images and also shows the coverage of the configuration space in the training data.

We train a CNN to predict the cart-position and pole-angle (cosine & sine of the angle) from images by minimizing an L2 loss over the dataset. The architecture of the CNN used is as follows:

```
[Conv(3x3, channels=32) → GroupNorm(groups=8) → ReLU → AvgPool(2x2) →
Conv(3x3, channels=64) → GroupNorm(groups=8) → ReLU → AvgPool(2x2) →
Conv(3x3, channels=128) → GroupNorm(groups=8) → ReLU → AvgPool(2x2) →
Flatten → Linear(32768, 256) → ReLU → Linear(256, 3)].
```

The state estimates (predicted cart position and pole angle) obtained by the CNN have a more realistic error profiles that are illustrated in Figure 20 over a separate test dataset of 20000 samples obtained by an uniform sampling of the configuration space. The nature of the error varies differently in the configuration space and can be thought of as providing a coarser sensing of the true state of the system.

The lookahead TD-MPC2 policy now operates on the current and previous state estimates i.e. the observation at time $t$ is $o_t = (\hat{y}_t, \hat{y}_{t-1})$. The agent predicts the force to apply on the cart to swing the pole upright and balance it – note that the instantaneous velocity information is not provided here.

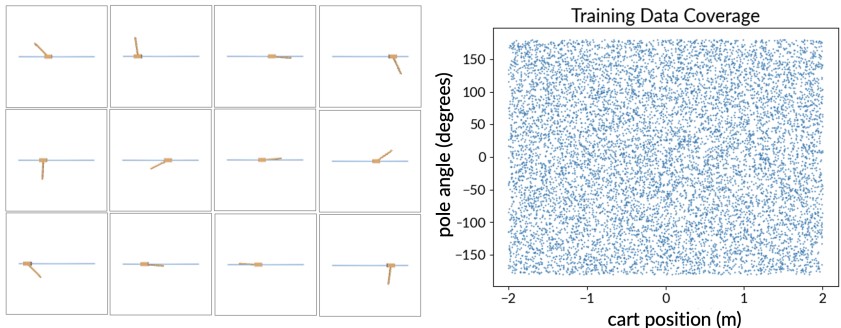

Figure 19: Sample images of the training dataset **(left)** and a scatter plot illustrating the coverage of the configuration space in the training data **(right)**.

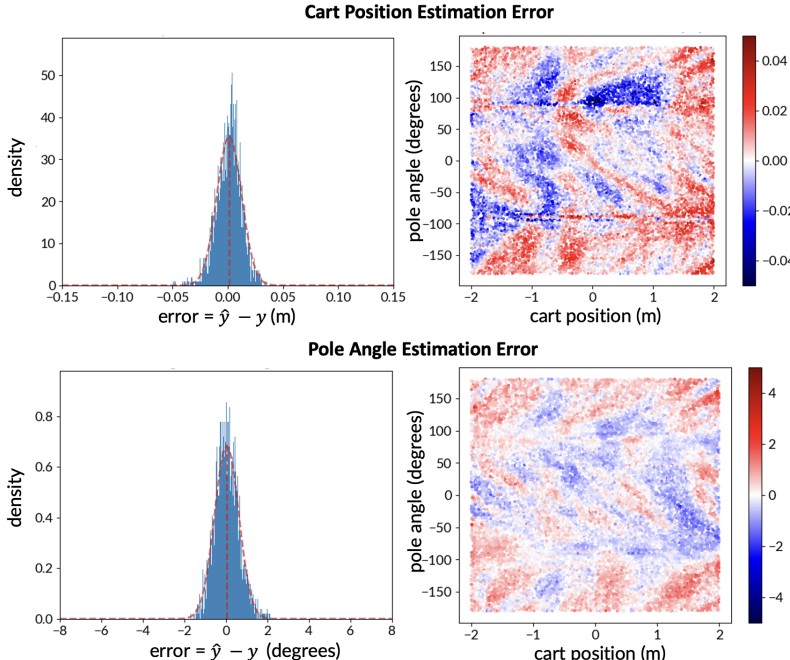

Figure 20: The error characteristics of the CNN based state estimator. The histogram on the left illustrates the distribution of errors (predicted - ground truth value) over states that are sampled uniformly from the configuration space. The plot on the right indicates the errors over the tested configuration space.

## I.2 VoSI Analysis

### Task Regret Profile Analysis

Like the analysis presented in Section 4 we start the investigation by analyzing the task regret profiles attained by fixed-rate mixed loop executions with varying execution period $h$ and obtain an approximate understanding of the relationship between sensing budget (as reflected in the sensing rate $\frac{1}{h}$) and task performance.

The agent with noisy sensing achieves a closed loop task performance of 150.45. To contrast the profile with "perfect sensing", we obtain the task-regret profile for another TD-MPC2 agent that is provided the true cart positions and pole angles for timesteps $t$ and $t - 1$. The agent under this "perfect sensing" setup achieves a closed loop performance of 164.46. However, do note that perfect sensing in this case does not accurately reflect the true Markov state of the system as the instantaneous velocities are not provided to the agent. The task regret profiles for both these sensing conditions are presented in Figure 21.

In Figure 21, we observe that with the fixed-rate mixed loop execution of the policy with noisy sensing any deviation from operating closed loop results in a performance drop. Whereas an agent provided with "perfect sensing" can tolerate a 10 fold reduction in the amount of sensing. However, bear in mind that fixed rate mixed loop execution is not necessarily the optimal sensing strategy and is merely a useful probe to arriving at an approximate understanding of the relationship between the sensing budget and task performance. We use VoSI profiles (introduced in Section 5) to provide a more fine-grained analysis of the agent at the state level by following the same protocol described in Appendix A.3 to gather data for the analysis.

### Characteristic VoSI profiles

In Figure 22 we visualize an overlay of VoSI profiles for states uniformly sampled from closed loop executions of the policy with noisy sensing. And similar to the insights of `swingup` task in the main paper we observe "stepped" VoSI profiles that involve a sharp increase in regret at some values $h$

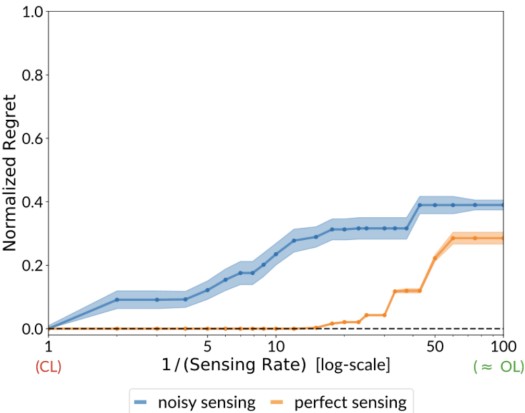

Figure 21: Task regret profiles for an agent with noisy sensing and perfect sensing on `swingup` task.

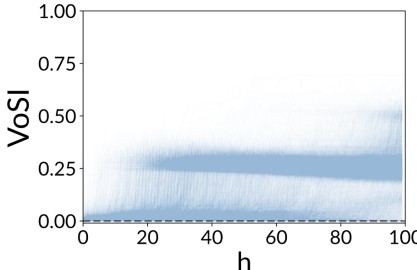

Figure 22: VoSI profiles overlayed for agent with noisy sensing (similar to Figure 9).

followed by steady increases or flat stretches, but these sharp increases occur at earlier open loop horizons $h$ compared to the perfect sensing scenario examined in the main paper (Figure 9-1).

The sharp increases in task regret primarily appear during the swingup phase of the task where not sensing during a short dynamic period of this phase can prevent the agent from correcting an undershoot which causes a sharp loss in rewards. For example, for states encountered in the 'swingup phase' the VoSI profiles (Figure 23-a, Figure 23-b) show a steep decrease in task performance when the agent forgoes sensing in a short window as extended execution of open loop actions based on noisy state estimate in this dynamic phase results in an undershoot of the pole (these are similar to insights we drawn from Figure 7-3a for an agent with perfect sensing). As the agent starts entering the 'balance phase' of the task (Figure 23-c → Figure 23-d → Figure 23-e) the agent starts to be able to tolerate extended horizons of open loop action execution without losing any task performance (similar to the insights obtained for agent with perfect sensing from Figure 7-3b, Figure 7-1d which show close to flat profiles during the 'balance phase' of the task).

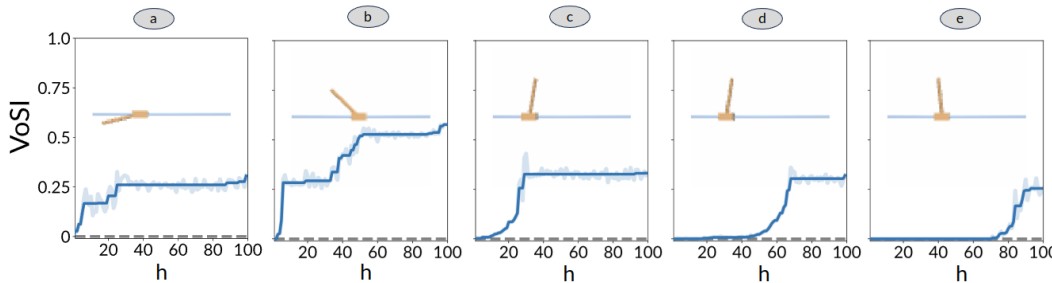

Figure 23: Characteristic VoSI profiles observed on the `swingup` task for an agent with noisy sensing.

## J  VoSI Analysis of a Complex Robotic Task: Quadruped Crate Climbing Task with Depth Camera and Proprioception

In this appendix, we look to demonstrate proof-of-concept VoSI application to still more complex and realistic robotic task: a Unitree Go1 quadruped performing a complex locomotion task, namely, climbing onto a crate. Aside from being more complex and realistic than the tasks studied thus far, this setting also facilitates the study of the value of *specific* sensors in a multi-sensory setup.

In particular, in many robotics applications an agent is typically equipped with a wide array of sensors that have different data acquisition and processing costs. For instance, our quadruped system comes equipped with joint encoders, force sensors on the feets, IMU and cameras to estimate the proprioceptive state and the exteroceptive terrain conditions. Of these sensory streams, constantly processing visual information from cameras poses significant challenges due to the limited on-board compute and power constraints of the system. In this section, we demonstrate how the proposed VoSI analysis can be used to characterize the value of *visual* sensing, assuming that other sensors are always available.

### Climb-Crate Task

In our task setup, a Unitree Go1 quadruped must climb onto a crate of varying heights using a head mounted depth camera (see Figure 24). For every episode, we spawn a crate with varying height (10 cm to 25 cm) while keeping its length and width fixed at 90 cm x 90 cm. The robot is spawned at a uniformly random distance (1m to 2m) away from the crate with slightly different approach angles ($\pm 10°$). The robot gets a forward velocity command that directs it to climb the crate, after it successfully climbs the crate the command is zeroed out.

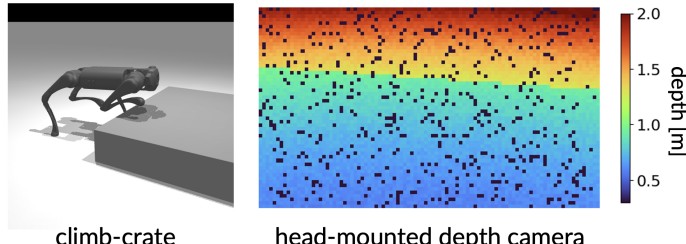

climb-crate          head-mounted depth camera

Figure 24: Illustration of the climb-crate task and the depth estimates from a head-mounted camera.

### Training

We leverage the widely used two-stage teacher-student training paradigm (Cheng et al., 2024; Kumar et al., 2021) to obtain a policy that can tolerate various degrees of staleness in the depth information while always receiving the latest proprioceptive information from the IMU and joint encoders.

During the RL training of the state-conditioned teacher policy we use dense rewards to encourage robust velocity-tracking walking gaits to emerge. Additionally, the teacher policy is trained on a wide range of terrain types with sensor noise and domain randomization. The teacher policy's behavior is then distilled to a student policy on the evaluated climb-crate task. During distillation, along with training a main closed loop policy network that operates on the latest sensory readings (depth and proprioceptive) to predict the teacher actions, we train an axillary network that operates on the history of proprioceptive readings and stale depth information to predict the same teacher actions. This allows the student to operate on stale depth information by choosing to query this auxiliary network. We adopt the architecture in Cheng et al. (2024) with the exception of dropping the recurrence in depth processing. The student policy operates on a 58x87 depth image with uniform random noise of $\pm 4$ cm in the depth estimates and random pixel dropout (dropout_p = 0.05), while reusing the proprioceptive noise levels and domain randomization parameters from Cheng et al. (2024). The trained student policy attains a closed loop success rate of 79.4% over 500 simulation trials on the climb-crate task. While we perform our analyses in simulation, these policies are known to transfer zero-shot to the real robot, so there is reason to believe that our analyses could continue to hold for the transferred policies.

## VoSI Analysis Results

Our VoSI analysis is applied to the depth sensing alone, while assuming that other sensors are always freely available. We closely follow the procedure outlined in Appendix A.3 where we first obtain about 50 closed loop rollouts and then conduct the VoSI analysis on subsampled states along these rollouts where the depth information is foregone to various extents.

Similar to the VoSI analysis conducted in previous sections, we analyze the effect of withholding the depth information to various extents as the agent performs the task and illustrate the characteristic profiles observed in Figure 25. Recall that VoSI evaluations are measured relative to a reward function: for simplicity, we use the sparse reward to compute VoSI, where the agent obtains a return of 1 if it successfully climbs onto the crate and gets 0 otherwise.

In episodes where the crate heights are high ($> 20$ cm) we observe that policy critically relies on the depth information moments before it has to climb the crate (Figure 25-A), but as soon as it has positioned one of it's feet on the crate the depth information does not add much value (Figure 25-B). Furthermore, in episodes where the crate height is lower $< 15$ cm the agent is able to successfully climb the crate without having access to the most recent depth information (Figure 25-C). This further highlights that the VoSI framework can provide useful insights to inform the deployment of such learned policies.

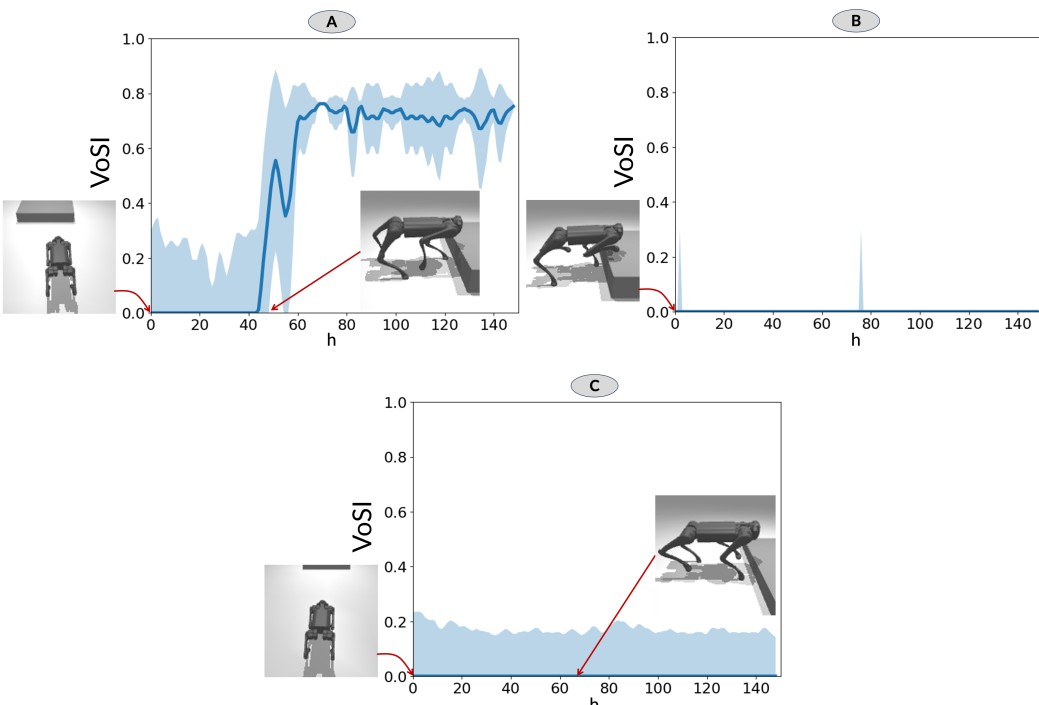

Figure 25: Characteristic VoSI profiles measuring extents to which the Go1 robot can operate without the latest depth information on the climb-crate task.

