# OpenReview forum: "The Value of Sensory Information to a Robot"
_ICLR.cc/2025/Conference — ICLR 2025 Poster_

### Official Review · Reviewer_Reqd · 2024-10-31

**Soundness:** 2
**Presentation:** 3
**Contribution:** 2
**Rating:** 6
**Confidence:** 3

**Summary:**

The paper investigates how frequently action sequences should be recomputed based on new observations with Diffusion Policy and TD-MPC2 on simulated benchmark tasks. The authors find that for many tasks, it is sufficient to recompute the action sequence at a low frequency, and for tasks from the Robosuite environment, it is even sufficient to use only the initial observations and execute entire rollouts open-loop. Furthermore, the authors identify stochasticity in the dynamics and modeling error as reasons why the action sequence needs to be updated based on new observations. They then devise a strategy for adapting the sensing and replanning frequency dynamically during the policy rollout.

**Strengths:**

The paper conducts an intriguing investigation of how frequent sensor input is needed during the execution of imitation / reinforcement learning policies. The paper is well-written, the experiments are clear, and the results are presented nicely. It is very interesting that the Robosuite tasks can be solved entirely with open-loop control, which suggests that this simulated benchmark might not capture the intricacies of real-world robotics applications well.

**Weaknesses:**

1. The paper aims to provide insights for robotics tasks, yet all results are obtained only in simulation. The characteristics of simulated benchmark are clearly very different from real-world robotics tasks. Simulations tend to behave significantly more deterministically and predictably. Real-world robotics tasks typically involve significant noise e.g., from the sensors and the actuation, and there are hard to model effects like friction and contacts, which are simplified significantly in dynamics simulators. The fact that the Robosuite benchmarks can be solved by only taking the initial observation into account further indicates that these benchmarks are too simplistic and do not capture the intricacies of real robot tasks. Furthermore, it seems challenging to conduct a similar analysis on real robots since the method requires collecting multiple rollouts for each state. I would appreciate it if the authors could discuss these limitations and the expected differences if such an analysis were applied for a real task.

2. It seems to me that the reward-free and interaction-free VoSI alternatives are still not very practical since these metrics cannot deal with situations in which there exist multiple possible strategies for solving a task. TD-MPC2 learns a dynamics model of the environment. Could we use the deviation between the rollouts predicted by the model and the true rollouts as a proxy for VoSI to circumvent the problem in this case?

3. The sensing frequency adaptation scheme was only tested on a gridworld task. Due to the discrete nature of the gridworld, the tested model is either perfectly accurate or predicting some completely different state. In robotics applications (the focus of the paper), models are typically always somewhat accurate and the main issue is accumulating errors. Clearly there are certain states/actions that cause larger model error (e.g., when the robot is in contact), but still this kind of model error seems to be quite different from the gridworld setting. In a robotics setting, I assume that the optimal sensing strategy is significantly closer to the fixed-rate strategy since preventing error accumulation requires frequent corrections. Perhaps the authors could comment on whether they expect similar benefits of the adaptation strategy in a robotics scenario or on how the experiment could be made more realistic.

4. I am unsure whether sensing frequency adaptation according to the VoSI can be useful for real-world robotics applications. In the current form calculating the VoSI requires multiple rollouts, which would defeat the purpose of increasing the computational efficiency. Furthermore, I am unsure whether a strategy to adapt the replanning frequency makes sense in typical real robotics applications. Usually the actions are executed at a fixed rate on the system, so the controller might be able to produce the action faster if it does not replan according to new sensor information, but this does not mean that the execution will be faster since then the controller just needs to wait according to the control frequency anyway. I would appreciate if the authors could expand on how this strategy could potentially be extended to real systems and what the benefits in practical applications could be.

5. Line 52: "Sensing only improves the agent actions significantly at some rare moments during task execution". This statement seems to hold only for certain tasks. Row 2 in Figure 7 shows that for at least two tasks, the performance degrades continuously the lower the sensing frequency is. If the statement above were accurate, then we would see a steep increase in the VoSI at the critical timestep (as e.g., in the swingup examples). I encourage the authors to revise this statement and acknowledge that the observation only holds for certain environments.

6. Line 290/291: The statement "agents [...] exposed to more training data usually degrade less" is not really backed by the data. This is only the case in in finger-spin, while in Push-T it seems to be the other way around. The authors should either provide more data to support the claim or revise the statement to make sure that it accurately reflects the data.

7. Line 810: The way that U(s') is defined, it is a uniform distribution over all states, which would mean that the model predicts that the agent teleports randomly across the gridworld. I feel like it should rather be a uniform distribution over something like the neighboring states as this seems like a more realistic error model (a realistic model would probably get the states at least approximately correct, but might make mistakes in the details).

8. There are relatively few tasks for which there are results for both Diffusion Policy and TD-MPC2 (only swingup and Push-T). It would be good to have results for more tasks with both methods to see how consistent the results are across the learning methodologies and policy architectures.

**Questions:**

1. Line 54: In my opinion it is not counter-intuitive that the agent's performance is inversely correlated with the sensor dependency, particularly in the case of TD-MPC2. TD-MPC2 learns a dynamics model of the environment that it uses for planning. The longer the agent was trained, the more accurate the model is, which means that the agent needs to correct the model error less frequently with real data. Perhaps the authors could expand on why they consider this inverse correlation as counter-intuitive.

2. In the plot for Push-T in Figure 10, it seems that most of the curves are quite flat, even though the contact dynamics should be relatively hard to predict for long horizons. Do all the flat curves correspond to cases where the T is almost at the target location or are there other situations in which the dynamics behave particularly predictable?

3. Figure 11: For "fixed-rate" the sensing rate specifies the frequency at which the policy gets new sensory input and recomputes the action sequence. For the "greedy" configuration there is no fixed rate as the execution strategy adapts the h dynamically. What exactly is plotted on the x-axis for the "greedy configuration"? Is it the average sensing frequency?

4. Line 835: If I understand correctly, this maximization is supposed to choose the largest h, so that the VoSI stays above the threshold. Shouldn't that rather be formulated as
$\mathrm{max}~ h ~~ \mathrm{s.t.}~ \mathrm{VoSI}(s_t, h) \leq h \delta$?

---

> ### Author Response · Authors · 2024-11-20
>
> We thank the reviewer for their detailed review!
>
> **The paper aims to provide insights for robotics tasks, yet all results are obtained only in simulation. The characteristics of simulated benchmarks are clearly very different from real-world robotics tasks..**
>
> We ask that you evaluate this paper not as introducing an immediately applicable engineering solution, but as the first empirical approach to address a core scientific question in robotics: “How critical is sensory information at each moment to a pre-trained policy?” See our [global response](https://openreview.net/forum?id=ikr5XomWHS&noteId=Y159uHUuRZ) above for further discussion on the relevance of our findings to real robotic tasks.
>
> **Furthermore, it seems challenging to conduct a similar analysis on real robots …**
>
> It is true that the VoSI measure defined in Eq(1) is not easy to implement on real robots. However, our interaction-free metrics (L473-502 + Appendix C) provide an alternative, often highly correlated method of implementing VoSI measurements, that we believe may be easier starting points for studies with real robots.
>
> **Reward-free and interaction-free VoSI alternatives are not very practical. In TD-MPC2, can we use deviation between predicted rollouts and true rollouts?**
>
> The plan disagreement metric proposed in the paper has a similar flavor to the reviewer’s proposal. In this metric we are measuring the deviation between the planned rollouts and the true rollout by computing the expected cumulative action disagreement. But, if the reviewer is suggesting capturing a notion of dynamics uncertainty, we believe TD-MPC2 directly does not support this investigation as the architecture assumes that the dynamics model is deterministic. Please let us know if we have misinterpreted your suggestion.
>
> **The sensing frequency adaptation scheme was only tested on a gridworld task. I assume optimal sensing strategy is significantly close to fixed-rate to periodically reduce error accumulation. … Unsure whether a strategy to adapt replanning frequency makes sense in typical real robotics applications.**
>
> We disagree that robotics environments would mainly only require fixed-rate sensing. The rate of accumulation of error is typically dramatically different during some phases of a task (e.g. contact-rich manipulation interactions or walking on rocky terrain) than during others (e.g. free space arm motion or walking on flat carpeted floors). For example, insights from our investigation also suggest similar characteristics in the finger-spin task: in the initial phase of the task the agent might need to sense more frequently (Figure 7-2b), but once the spinner is already spinning above a desired velocity the sensing rate can be lower (Figure 7-2c).
>
> On the relevance of adaptive replanning, many prior works have indeed pursued this line of thinking for computational efficiency in real practical systems that either have real-time requirements or operate with limited computational resources. For instance, in distributed networked systems where the communication layer imposes bandwidth and latency constraints – knowing when information is valuable to process and transmit for replanning can be very beneficial to characterize for deployment of these policies.
>
> Finally, note that the sensing frequency adaptation shown in Fig 11 is not the main focus of the paper. Our algorithm there is a naive greedy approach that only scales to small environments like grid-world. This is only intended as a proof-of-concept.

---

> > ### Author Response · Authors · 2024-11-20
> >
> > **Line 52: "Sensing only improves the agent actions significantly at some rare moments during task execution". This statement seems to hold only for certain tasks. Row 2 in Figure 7 shows that for at least two tasks, the performance degrades continuously the lower the sensing frequency is. … I encourage the authors to revise this statement and acknowledge that the observation only holds for certain environments.**
> >
> > No, Figure 7 does not provide any plots that summarize VoSI for a task, or even individual trajectories in a plot. Instead, it shows some prototypical VoSI profile shapes. Each plot in Fig 7 corresponds to a single last-observed state $s_t$, and how much loss in task value occurs over time $>t$ when executing an open-loop starting from that state. As such gradual degradation profiles in Figure 7, Row 2 indicate that the loss in sensory information leads to steadily increasing value loss, but it is not sufficient to conclude that it would have been valuable to sense at each moment in that window. For example, Figure (9,16): show VoSI profiles over the course of a rollout and is suggestive of the fact sensing at a few moments over the rollout might suffice to retain _much_ of the closed loop performance. However, we agree with the reviewer that such statements are very much task-dependent and depend on how much performance loss is acceptable and we will revise it to say “In most tasks”. Also, our description in Lines 371-373 was a potential source of confusion here, and we have now improved it, thank you for the question.
> >
> > **Line 290/291: The statement "agents [...] exposed to more training data usually degrade less" is not really backed by the data. This is only the case in finger-spin, while in Push-T it seems to be the other way around.**
> >
> > Thank you for this question, the plots in Fig 5 are a bit hard to read, and also don’t tell the full story. We will clarify now.
> >
> > Fig 5 (or it’s zoomed counterpart in Fig 18 (Appendix G.1)) shows a drop in performance of mixed-loop execution relative to closed-loop performance for each policy, but closed-loop performance for beginner and novice are often poor, meaning that there is “less to lose” when executing mixed-loop. For example, in the Push-T task, our “beginner” policy executed closed-loop gets a return of 25.62 compared to 66.59 for the expert. Thus, even if “beginner” deteriorates to a lower return with mixed-loop execution, this registers in the “relative regret” performance metric as a relatively small drop in task regret.
> >
> > In Appendix G.3 (Figure 21), we provide a version of the plot where the regret scales are the same by computing the task performance regret with respect to the performance of the “expert” policy checkpoint for training stage (and “large” checkpoint for the model capacity experiment). From Figure 21, it becomes more evident that the expert is capable of retaining higher levels of performance as sensing rate decreases.
> >
> > **Line 810: The way that U(s') is defined … would mean that the model predicts that the agent teleports randomly across the gridworld. I feel like it should rather be a uniform distribution over something like the neighboring states as this seems like a more realistic error model**
> >
> > We thank the reviewer for this suggestion and we agree with the reviewer that the chosen error model does not reasonably represent transition dynamics error observed in practice. We have fixed this now and use the proposed more realistic error model where the agent sometimes moves to a wrong but neighboring cell, rather than teleport it to any cell over the full grid. We have redone the four-rooms experiments with this change and updated plots are provided in Appendix G.2 (Figure 19 and Figure 20). we observe that this change does not impact the conclusions we draw from these experiments i.e. (a) the discussion on how dynamics error necessitates sensing (b) our proposed greedy strategy based on VoSI ($\eta_\text{greedy}$) can effectively exploit VoSI profiles to achieve better sensing reduction performance than fixed rate baselines, and the gap is even wider than in our earlier results. We will incorporate these changes into the main paper.
> >
> > **There are relatively few tasks for which there are results for both Diffusion Policy and TD-MPC2 (only swingup and Push-T).**
> >
> > As we discussed in L194, we did not report results on more tasks as we failed to obtain performant models to do a comparable analysis, in Appendix A.2 we discuss a hypothesis for why we failed to obtain effective Diffusion Policy controllers on the finger-spin and cup-catch tasks. The comparison of two different policy synthesis approaches on the same task is a very interesting question, which requires further systematic investigation. On the tasks where we obtained results for the two we observed only some second order effects (discussed in Appendix D) with limited significance for the main results reported in the paper.

---

> > > ### Author Response · Authors · 2024-11-20
> > >
> > > **Line 54: In my opinion it is not counter-intuitive that the agent's performance is inversely correlated with the sensor dependency, particularly in the case of TD-MPC2. The longer the agent was trained, the more accurate the model is, which means that the agent needs to correct the model error less frequently with real data.**
> > >
> > > This is indeed the explanation we provide in Sec 4.1, and we of course hope that the explanation alleviates the counter-intuitiveness.
> > >
> > > We believe however that many readers might find that result counter-intuitive upon first reading. On its surface, it is easy to agree with a statement like: “if you cannot respond to environment stimulus, your policy is likely to perform poorly”. After all, this is why we expect closed loop controllers in general to outperform open loop ones. However, in our setting, we find that well-trained controllers approach open-loop-like behaviors i.e. they are least dependent on sensing. This finding may at first appear to go against the above intuition.
> > >
> > > **In the plot for Push-T in Figure 10, it seems that most of the curves are quite flat, even though the contact dynamics should be relatively hard to predict for long horizons. Do all the flat curves correspond to cases where the T is almost at the target location or are there other situations in which the dynamics behave particularly predictable?**
> > >
> > > Thank you for pointing this out. We were curious about this ourselves, and upon examination, the explanation is simple. The plots in Fig 10 overlay VoSI profiles sampled uniformly from all states from episodes of fixed length. In many of these episodes, the task objective is achieved much before the episode ends i.e. the T object is correctly aligned with the goal. At this point, VoSI profiles are flat because actions are trivial and the policy does not perform further meaningful actions. This is why there is a high density of seemingly flat profiles in Fig 10. We will add this explanation to the main paper or appendix.
> > >
> > > **Figure 11: For the "greedy" configuration there is no fixed rate as the execution strategy adapts the h dynamically. What exactly is plotted on the x-axis for the "greedy configuration"? Is it the average sensing frequency?**
> > >
> > > Yes, The sensing rate for the “greedy” dynamic sensory-policy is the average fraction of timesteps at which the agent acquired sensory information. We will clarify this in the text or appendix of the paper.
> > >
> > > **Line 835: If I understand correctly, this maximization is supposed to choose the largest h, so that the VoSI stays above the threshold. Shouldn't that rather be formulated as max $h$  s.t. $\text{VoSI}(s_t, h) \leq h \delta$**
> > >
> > > This is exactly right, and also what we mean by the notation in the algorithm: $\mathop{argmax}_h \text{VoSI}(s_t, h) ≤ h · \delta$

---

> > ### Comment · Reviewer_Reqd · 2024-11-22
> >
> > > We ask that you evaluate this paper not as introducing an immediately applicable engineering solution, but as the first empirical approach to address a core scientific question in robotics: “How critical is sensory information at each moment to a pre-trained policy?”
> >
> > I am aware that a single paper does not immediately have to solve all the practical problems and provide a perfect solution for real robotics tasks. Yet, significant differences between the setting that is used as motivation (real robotics tasks) and the setting that is investigated (simulation and gridworlds) have to be counted as a limitation of the paper. However, it is not my intention to discard your simulation results as irrelevant, but it would be very helpful to discuss in which way the results potentially differ if a similar analysis were applied to a real robot learning task.
> >
> > > It is true that the VoSI measure defined in Eq(1) is not easy to implement on real robots. However, our interaction-free metrics (L473-502 + Appendix C) provide an alternative, often highly correlated method of implementing VoSI measurements, that we believe may be easier starting points for studies with real robots.
> >
> > As mentioned in point 2. in the review, the interaction-free metrics do not seem to be suited for situations in which there exist multiple potential solutions (which is very common in realistic tasks). I would appreciate if the author could discuss potential directions for circumventing these limitations.
> >
> > > The plan disagreement metric proposed in the paper has a similar flavor to the reviewer’s proposal. In this metric we are measuring the deviation between the planned rollouts and the true rollout by computing the expected cumulative action disagreement. But, if the reviewer is suggesting capturing a notion of dynamics uncertainty, we believe TD-MPC2 directly does not support this investigation as the architecture assumes that the dynamics model is deterministic. Please let us know if we have misinterpreted your suggestion.
> >
> > What I meant was to execute an action sequence, then take that action sequence and use TD-MPC2's model to predict the same trajectory by feeding the initial state and the action sequence into the model and then compare the predicted trajectory to the real trajectory. That way, the model only needs to produce a deterministic predcition of the resulting states and no uncertainty estimate. Would such a strategy yield a sensible VoSI estimate?
> >
> > > We disagree that robotics environments would mainly only require fixed-rate sensing. The rate of accumulation of error is typically dramatically different during some phases of a task (e.g. contact-rich manipulation interactions or walking on rocky terrain) than during others (e.g. free space arm motion or walking on flat carpeted floors).
> >
> > I agree that the scale of error is quite different depending on the context of the robot and I do not think that fixed-rate sensing is necessarily the optimal strategy for these tasks. Yet, I think in most robotics tasks the main problem is still an accumulation of error. E.g., in nonprehensile manipulation, the object might move slightly different than expected due to friction at every step. This is very different from the sudden discrete errors for the gridworld tasks, where the sensing frequency adaptation was evaluated.
> >
> > > On the relevance of adaptive replanning, many prior works have indeed pursued this line of thinking for computational efficiency in real practical systems that either have real-time requirements or operate with limited computational resources. ...
> >
> > This example is still not very clear to me. I assume in most robotics applications the bandwidth / latency of the communication suitable to provide sensor readings at a fixed frequency. In which scenarios would it be beneficial to skip reading sensor measurements in some steps?
> >
> > > No, Figure 7 does not provide any plots that summarize VoSI for a task, or even individual trajectories in a plot. ...
> >
> > Thank you for the clarification. I appreciate the revision of the statement in line 52.
> >
> > > Thank you for this question, the plots in Fig 5 are a bit hard to read, and also don’t tell the full story. ...
> >
> > I still do not think that the statement "agents [...] exposed to more training data usually degrade less at lower sensing rates" is reflected in the data. The only sensible metric for performance degradation is the drop in performance relative to the policy's closed-loop performance (as in Figure 5). I am aware that the beginner policy has less performance to lose, so in that sense the metric is not perfect, but still I do not see enough evidence in both Figures 5 and 21 that would support the conclusion above.

---

> > > ### Comment · Reviewer_Reqd · 2024-11-22
> > >
> > > > We thank the reviewer for this suggestion and we agree with the reviewer that the chosen error model does not reasonably represent transition dynamics error observed in practice. ...
> > >
> > > Thank you for this additional experiment. The results are indeed quite similar and the experiment is more convincing with this updated noise model.
> > >
> > > > Thank you for pointing this out. We were curious about this ourselves, and upon examination, the explanation is simple. ...
> > >
> > > Thanks for this explanation. Please add this clarification to the paper. It could also make sense to plot these trivial trajectories where the agent already succeeded in a different color or filter them out entirely (also for cup-catch).
> > >
> > > > Yes, The sensing rate for the “greedy” dynamic sensory-policy is the average fraction of timesteps at which the agent acquired sensory information. ...
> > >
> > > Thank you for the clarification.
> > >
> > > > This is exactly right, and also what we mean by the notation in the algorithm: $\mathop{argmax}_h \text{VoSI}(s_t, h) ≤ h · \delta$.
> > >
> > > This notation seems non-standard and had me confused a bit. Please consider revising it.

---

> > > > ### Author Response · Authors · 2024-11-27
> > > >
> > > > We thank the reviewer for their response. We have now incorporated the promised changes and results of the experiments proposed into the main paper as outlined in the [global response](https://openreview.net/forum?id=ikr5XomWHS&noteId=rvYibL0A1s). We address the remaining concerns below:
> > > >
> > > > > **Significant differences between the setting that is used as motivation (real robotics tasks) and the setting that is investigated (simulation and gridworlds) have to be counted as a limitation of the paper. … It would be very helpful to discuss in which way the results potentially differ if a similar analysis were applied to a real robot learning task.**
> > > >
> > > > We thank the reviewer for the comment. In our discussion and conclusion section (L523-539) of the paper, we do list this as a limitation of the study and have mentioned that our contribution here is just the first step towards the ambitious goal of systematically analyzing robotic tasks that fully reflect the complexities of the real world.
> > > >
> > > > As we state in our [earlier response](https://openreview.net/forum?id=ikr5XomWHS&noteId=UupSBwk9cN), the VoSI measure as defined in Eq(1) is not easy to implement on real robots. This was our inspiration to study interaction-free metrics (as mentioned in L473-477). Our primary examination of some interaction-free metrics yielded signs of positive correlation with the original metric in Eq (1) but we have not yet studied them on real robots in this manuscript. Finally, our findings in simulation are already partly mirrored in previous findings on real robot tasks: there is growing evidence from real robot learning experiments that open loop baselines can indeed be effective in several table-top manipulation and locomotion tasks, similar to our findings on Robosuite simulated tasks (as mentioned in L191-192).
> > > >
> > > > > **The proposed interaction-free metrics do not seem to be suited for situations in which there exist multiple potential solutions (which is very common in realistic tasks). … could authors discuss potential directions for circumventing these limitations.**
> > > >
> > > > We don’t think this is true: our state-disagreement-based metric [L914-917] does in fact account for multimodality in the original policy: we first generate multiple closed-loop executions to capture multiple modes if any. Then, we generate multiple open-loop executions, and compare each open-loop execution to its closest closed-loop execution, a kind of Chamfer distance between the two sets of executions.
> > > >
> > > > Please recall also that we are studying the value of the sensory information in the context of a pre-trained policy, so the only multimodality we are concerned with is multimodality in the closed-loop executions of that policy, as handled above.
> > > >
> > > > > **Suggestion: Execute an action sequence, then take that action sequence and use TD-MPC2's model to predict the same trajectory by feeding the initial state and the action sequence into the model and then compare the predicted trajectory to the real trajectory. Would such a strategy yield a sensible VoSI estimate?**
> > > >
> > > > Sorry, we might have misunderstood this suggestion earlier, and thank you for the clarification. However, we do not believe this is an appropriate strategy to measure VoSI: it only measures whether the learned dynamics model is good at predicting dynamics under the closed-loop actions, but does not directly measure what would happen if sensory inputs were withheld from the agent for some interval i.e., the mixed-loop term in the original VoSI definition of Eq (1).  .
> > > >
> > > > Our plan-disagreement metric (L918-921) is more directly connected to Equation (1), and can also be computed in the interaction+reward-free setting. Further, our metric also has the advantage that it does not only apply to methods that learn an explicit model of the dynamics. This permits us to analyze other policy classes such as diffusion policies.
> > > >
> > > > Your suggested approach to measure the fidelity of dynamics on closed loop actions is closely connected to our reasoning in L213-245 in the paper: as stated there, it measures the information content in sensory observations broadly, but not task-relevant information content.

---

> > > > > ### Comment · Reviewer_Reqd · 2024-11-29
> > > > >
> > > > > > In our discussion and conclusion section (L523-539) of the paper, we do list this as a limitation of the study and have mentioned that our contribution here is just the first step towards the ambitious goal of systematically analyzing robotic tasks that fully reflect the complexities of the real world.
> > > > >
> > > > > Yes, I am aware of the challenges of applying such an analysis to real robotics tasks. I was just wondering whether you have any hypotheses how the analysis results would change if it were possible to conduct the analysis for such tasks. The noisy sensing setting in appendix I is already a first step in the direction of realistic tasks, so thanks for these additional experiments.
> > > > >
> > > > >
> > > > > > We don’t think this is true: our state-disagreement-based metric [L914-917] does in fact account for multimodality in the original policy: we first generate multiple closed-loop executions to capture multiple modes if any. Then, we generate multiple open-loop executions, and compare each open-loop execution to its closest closed-loop execution, a kind of Chamfer distance between the two sets of executions.
> > > > >
> > > > > I was not aware that the state-disagreement metric searches for the most similar closed-loop trajectory first, so thanks for the clarification. This should already help alleviate some of the issues related to stochasticity in the policy. However, this does not seem to be enough if the policy is very stochastic, as in the case of the degenerate phases mentioned in lines 484-501 (in that case, the policy might choose any action sequence). Do you see a way to address this issue?
> > > > >
> > > > > For the plan-disagreement metric a similar problem arises. Both diffusion policy and the CEM planner of TD-MPC2 are stochastic, so the look-ahead action sequences might deviate from the closed-loop action sequence simply due to sampling. The computed look-ahead action sequences might still be valid strategies that solve the task, but if they deviate from the closed-loop action sequence, this will result in a low VoSI-P score. Do you see a potential direction to address this issue for the VoSI-P metric?
> > > > >
> > > > >
> > > > > > However, we do not believe this is an appropriate strategy to measure VoSI: it only measures whether the learned dynamics model is good at predicting dynamics under the closed-loop actions, but does not directly measure what would happen if sensory inputs were withheld from the agent for some interval i.e., the mixed-loop term in the original VoSI definition of Eq (1). ...
> > > > >
> > > > > TD-MPC2 uses MPC with the planned model, so there should be a strong correlation between the model error and the frequency at which the agent should replan. I am aware that this method is not applicable to model-free methods like diffusion policy, but in my opinion it would be interesting to see whether this simple-to-compute quantity (it essentially requires just a single model rollout), could be used for a similar purpose as the more expensive VoSI measures (at least for planning-based methods). Also such a metric should be able to address concerns with stochasticity in the policy.
> > > > >
> > > > >
> > > > > > Error does indeed accumulate, but not at the same rate, and there are jumps and discontinuities in how it accumulates. ...
> > > > >
> > > > > Thank you for these additional examples. For these kinds of tasks, I can imagine that the optimal sensing strategy might be very different from fixed-rate sensing.
> > > > >
> > > > >
> > > > > > The setup in our previous response exemplifies the kind of problem studied in the networked control systems literature. ...
> > > > >
> > > > > Thank you for the clarification; it has made the motivation behind your work clearer to me.
> > > > >
> > > > >
> > > > > > We will modify this claim to say – “appear to” as the claim clearly holds for the finger-spin task but cannot be crisply captured by a single metric in the push-T task. Relative drop as a performance metric is indicative but not the final word: it requires some interpretation in the context of the specific reward function for each task, and the baseline rewards of the closed-loop controller relative to random behavior etc. We have provided more context near the results in Fig 5 to reflect this.
> > > > >
> > > > > In my opinion the claim is still formulated a bit to general, given that there is really only a single datapoint for which we can observe this phenonmeon (finger-spin). For measuring "performance degradation", the only relevant metric should be the one of Figure 5. If the comparison is flawed for Push-T, Figure 5 might not be a definite counter-argument for this statement, but still there is - in my eyes - no supporting evidence in the case of Push-T for this statement. Please consider either stating explicitly that you observed this phenomenon for finger-spin or changing the claim to reflect more accurately the observations that you have made on more than one task.
> > > > >
> > > > >
> > > > > > Thank you for the suggestion. We have fixed this in the revised manuscript.
> > > > >
> > > > > Thank you, now the notation seems clearer to me.

---

> > > > > > ### Author Response · Authors · 2024-12-01
> > > > > >
> > > > > > Thank you for continuing to engage with us and offer valuable feedback.
> > > > > >
> > > > > > > **Do you have any hypotheses how the results would change if it were possible to conduct the analysis on real robotic tasks? New experiments already a first step.**
> > > > > >
> > > > > > Yes, we believe that our results on simulated tasks have a high chance of being reflective of results on real robotic tasks. In the original submission, we had argued (now in Lines 262-296, Fig 4, Fig 5) that finite model capacity can have similar effects on the value of sensing as true environment stochasticity, and this was supported by experiments within our simple gridworld environment. Further, we had argued that our findings regarding the effectiveness of open-loop policies on Robosuite manipulation tasks are also backed by previously reported results on real robots (now in Lines 188-193). As stated in your comment, our [new results](https://openreview.net/forum?id=ikr5XomWHS&noteId=rvYibL0A1s) on noisy sensing in the swingup task add further evidence that the findings on simulated tasks already offer useful intuitions for more realistic settings such as real robotic tasks.
> > > > > >
> > > > > > > **I am aware that suggested dynamics-based metric is not applicable to model-free methods like diffusion policy, but it would be interesting to see whether this quantity (it essentially requires just a single model rollout), could be used for a similar purpose**
> > > > > >
> > > > > > Thank you for suggesting this third alternative approach to implementing VoSI. We have now implemented it, and have added it to our paper as “VoSI*-M”, for Model error-based VoSI, and an asterisk to indicate that it does not involve any actual open-loop executions.
> > > > > >
> > > > > > Since VoSI*-M only applies to model-based methods, we applied this to TD-MPC2 policies alone. Further, TD-MPC2 predicts dynamics in a learned latent space, so we compute VoSI*-M in the learned latent space, as follows:
> > > > > >
> > > > > > $$\text{VoSI*-M}(s, h) = E_{a_{0:h} \sim \tau_{\text{CL}}(s)} \left[ || \hat{z}_h - z_h ||_2    \right]$$
> > > > > >
> > > > > > Where $z_h = \text{Encode}(s_h)$ ; $\hat{z}_t = f\(\hat{z}\_{t-1}, a\_{t-1}\)$ for $t > 0$ with $\hat{z}\_0 = \text{Encode}(s\_0)$ – denoting repeated application of the dynamics with action sequence $a\_{0:h}$ starting the state $s = s\_0$. Note that VoSI*-M also requires estimating an expectation over closed-loop executions, so it involves multiple samples of the closed-loop distribution $\tau\_{\text{CL}}(s)$, followed by one model rollout per closed-loop sample. Closed-loop executions in the actual environment are typically more expensive than model rollouts, and this is especially true for methods like TD-MPC2 whose model rollouts happen directly in a compact latent space.
> > > > > >
> > > > > > We have implemented this for TD-MPC2 on tasks swingup, finger-spin, and cup-catch from the dm-control suite. As with our previous VoSI alternatives (VoSI-P and VoSI-S), we test VoSI*-M’s rank order correlation with the original return degradation-based VoSI metric (Eq 1, VoSI-R). as done in the main paper (L480-484) and report the result from Table 1 below (with a new row):
> > > > > >
> > > > > > | correlation | swingup | cup-catch | finger-spin |
> > > > > > |---|---|---|---|
> > > > > > | $\rho(\text{VoSI-R}, \text{VoSI-S})$ | 0.78 | 0.88 | 0.98 |
> > > > > > | $\rho(\text{VoSI-R}, \text{VoSI-P})$ | 0.54 | 0.18 | 0.73 |
> > > > > > | $\rho(\text{VoSI-R}, \textbf{VoSI*-M})$ | 0.89 | 0.79 | 0.74 |
> > > > > >
> > > > > > VoSI*-M appears to be a suitable interaction-free estimate of VoSI: it is better than VoSI-P particularly on swingup and cup-catch tasks. **We will add these results to Table 1**. This might indeed prove to be a useful VOSI estimate for model-based policies on real-robot tasks, thank you very much for suggesting it and pushing us to add this experiment!
> > > > > >
> > > > > > > **already help alleviate some issues related to stochasticity, but what if the policy is very stochastic?**
> > > > > >
> > > > > > The Chamfer distance approach we described above does not in theory break down in such cases, but they can indeed cause practical issues. In general, the more the stochasticity in the policy, the more computationally expensive it is to get a good estimate of our VoSI scores, because we would need more samples of closed-loop and mixed/open-loop executions to get representative measurements.  Future work could quantify how many rollouts will be required to obtain reasonable estimates of the metric given an empirical notion of the learned policy’s variance.
> > > > > >
> > > > > > Note that this problem does not completely go away for your suggested dynamics-based metric (VoSI*-M): if the closed-loop policy is highly stochastic, it would be necessary to get many closed-loop executions as pointed out above. But yes, VoSI*-M has the advantage that it does not also require mixed-loop or open-loop rollouts online like VoSI-R/S.

---

> > > > > > > ### Comment · Reviewer_Reqd · 2024-12-02
> > > > > > >
> > > > > > > > Yes, we believe that our results on simulated tasks have a high chance of being reflective of results on real robotic tasks. In the original submission, we had argued (now in Lines 262-296, Fig 4, Fig 5) that finite model capacity can have similar effects on the value of sensing as true environment stochasticity, and this was supported by experiments within our simple gridworld environment. ...
> > > > > > >
> > > > > > > Thank you for this discussion. There are some further effects like delays, partial observability, and hard-to-predict friction that might lead to different results on real systems, but this paper is already a step in the right direction. Perhaps, these additional effects could be studied in depth in future work.
> > > > > > >
> > > > > > >
> > > > > > > > Thank you for suggesting this third alternative approach to implementing VoSI. We have now implemented it, and have added it to our paper as “VoSI*-M”, for Model error-based VoSI, and an asterisk to indicate that it does not involve any actual open-loop executions. ...
> > > > > > >
> > > > > > > Thank you for adding this very insightful experiment. I believe VoSI-M to be a useful and lightweight metric for estimating VoSI in the context of planning-based methods. The results look very promising.
> > > > > > >
> > > > > > >
> > > > > > > >  Future work could quantify how many rollouts will be required to obtain reasonable estimates of the metric given an empirical notion of the learned policy’s variance.
> > > > > > >
> > > > > > > Yes, this sounds like a very sensible direction for future work that has the potential to facilitate such an analysis on real-world tasks.
> > > > > > >
> > > > > > >
> > > > > > > > Note that this problem does not completely go away for your suggested dynamics-based metric (VoSI*-M): if the closed-loop policy is highly stochastic, it would be necessary to get many closed-loop executions as pointed out above. But yes, VoSI*-M has the advantage that it does not also require mixed-loop or open-loop rollouts online like VoSI-R/S.
> > > > > > >
> > > > > > > That is true, it does not entirely avoid the problem, but I think removing the need for additional mixed-loop rollouts is already quite valuable.
> > > > > > >
> > > > > > >
> > > > > > > > So we will revise Line 291 to say: “In line with the arguments above, agents with high model-capacity have lower normalized regret curves, i.e. they degrade less at lower sensing rate. This is also true for agents trained with more data, particularly on finger-spin. Push-T training data results are less conclusive (see Appendix G).”
> > > > > > >
> > > > > > > Thanks for revising this claim. Now it seems accurate to me.
> > > > > > >
> > > > > > >
> > > > > > > > Thank you once again for continuing to invest your effort into improving our paper. We believe our paper has grown significantly stronger through the process of responding to your comments. Please let us know if any concerns remain.
> > > > > > >
> > > > > > > Thank you for the time and effort for all these clarifications and additional experiments. They helped me understand the results better and have clearly made the paper stronger. I have no further questions and I increased my score.

---

> ### Author Response · Authors · 2024-11-27
>
> > **Yet, I think in most robotics tasks the main problem is still an accumulation of error ... the object might move slightly different than expected due to friction at every step. This is very different from the sudden discrete errors for the gridworld tasks …**
>
> Error does indeed accumulate, but not at the same rate, and there are jumps and discontinuities in how it accumulates. Consider a spinning cube dropped onto the floor: the contact event when it makes contact with the floor can lead to many wildly divergent outcomes arising from minute differences in the configuration at the time of contact. Contact-rich tasks involving repeated making and breaking of contacts are difficult precisely because of these phenomena. This is connected to our examples in our previous response — the free space motion of an arm usually incurs very little accumulation of error before and after a grasp, but it is very difficult to predict exactly how an object such as a spoon might be oriented in the gripper after the grasp – all the error accumulation happens in the grasping phase, and it is more important to frequently sense the world during that phase (if the object orientation is important to get right for the task).
>
> > **This example is still not very clear to me. I assume in most robotics applications the bandwidth / latency of the communication suitable to provide sensor readings at a fixed frequency. In which scenarios would it be beneficial to skip reading sensor measurements in some steps?**
>
> Sorry, we should have cited the literature that studies these problems, and perhaps it would have been clearer. Thank you for checking back with us to give us an opportunity to clarify. The setup in our previous response exemplifies the kind of problem studied in the networked control systems literature [1-3]. We explain more clearly here: In networked control systems often the communication layer imposes severe bandwidth and latency constraints, e.g.: a marine robot/robots operating in remote locations[1], in such scenarios knowing when information is valuable to transmit and process can be quite beneficial to minimize the overall communication cost[2,3].
>
> However, networked control systems only help to make very obvious why skipping sensing is of interest. Even in a single-agent setting, if the agent has limited computational resources, it is often beneficial to avoid processing the sensory inputs and replanning based on them, even when the sensory inputs themselves are available at each moment in time. This is in fact common practice today with lookahead policies such as diffusion policies and action-chunking transformers that require non-trivial computation and time, even when they are typically run on well-equipped desktop computers connected to a robot. On more constrained robots (e.g. imagine a small UAV robot operating with onboard compute), these constraints are even more real, even for simpler policies.
>
> > **I still do not think that the statement "agents [...] exposed to more training data usually degrade less at lower sensing rates" is reflected in the data.**
>
> We will modify this claim to say – “appear to” as the claim clearly holds for the finger-spin task but cannot be crisply captured by a single metric in the push-T task. Relative drop as a performance metric is indicative but not the final word: it requires some interpretation in the context of the specific reward function for each task, and the baseline rewards of the closed-loop controller relative to random behavior etc. We have provided more context near the results in Fig 5 to reflect this.
>
> > **This notation seems non-standard and had me confused a bit. Please consider revising it.**
>
> Thank you for the suggestion. We have fixed this in the revised manuscript.
>
> **References**
>
> [1] Zolich, Artur, et al. "Survey on communication and networks for autonomous marine systems." Journal of Intelligent & Robotic Systems 95 (2019): 789-813.
>
> [2] Solowjow, Friedrich, et al. "Event-triggered learning for resource-efficient networked control." 2018 Annual American Control Conference (ACC). IEEE, 2018.
>
> [3] Kepler, Michael E., and Daniel J. Stilwell. "An approach to reduce communication for multi-agent mapping applications." 2020 IEEE/RSJ International Conference on Intelligent Robots and Systems (IROS). IEEE, 2020.

---

> ### Author Response · Authors · 2024-12-01
>
> > **there is - in my eyes - no supporting evidence in the case of Push-T for this statement (more training data leads to lower sensitivity to sensing rates). Please consider either stating explicitly that you observed this phenomenon (only) for finger-spin or changing the claim to reflect more accurately the observations that you have made on more than one task.**
>
> This is fair, we can no longer edit the paper, but we will add qualitative evidence (policy rollout visualizations) to Appendix G in our next revision / camera-ready that will add further evidence for this statement in Push-T. However, we agree overall that the evidence for Push-T is not as clear-cut as for finger-spin. So we will revise Line 291 to say: “In line with the arguments above, agents with high model-capacity have lower normalized regret curves, i.e. they degrade less at lower sensing rate. This is also true for agents trained with more data, particularly on finger-spin. Push-T training data results are less conclusive (see Appendix G).”
>
>
> Thank you once again for continuing to invest your effort into improving our paper. We believe our paper has grown significantly stronger through the process of responding to your comments. Please let us know if any concerns remain.

---

### Official Review · Reviewer_6eFu · 2024-10-31

**Soundness:** 3
**Presentation:** 2
**Contribution:** 1
**Rating:** 5
**Confidence:** 5

**Summary:**

The paper empirically studies the value of sensory information in a variety of robotic tasks and for policies stemming from different learning approaches. This is achieved by mixing open-loop and closed-loop control. Performance degradation is analyzed and its correlations with sensing-frequency, task proficiency, task difficulty, etc. are analyzed. VoSI is proposed as a metric, as well as derived variants that are reward-free and interaction-free. Finally the paper shows a first attempt at using the VoSI profiles for control with parsimonious sensing. The paper studies 7 robotic tasks and a benchmark, which reveal insights.

**Strengths:**

The paper is well written and tackles an interesting and understudied problem, that has practical relevance in robotics.
The extensive experiments are well designed and executed. Having a diverse set of tasks and environments is appreciated.
The methodology for analysis is sound, the proposed metrics potentially more widely applicable, and the final proposal for using them for control interesting.
The paper is well written and easy to follow. The supplementary material is well done and provides many additional relevant details.

**Weaknesses:**

- In my opinion the paper studies entirely the wrong setting
  - It only becomes clear rather late in the paper that all experiments (except 4-room) are done in a setting assuming perfect sensing and that everything is deterministic. That's exactly the opposite of how things work on a real robot and where the challenges lie. Sensing is going to be imperfect, and a robot has to deal with unknown perturbations, changes and variability in the environment/objects/tasks, etc. The authors mention those things as future work, but I'd argue that without considering those the paper isn't much more than an academic exercise in a setting that is simplified to a point it doesn't have any practical use for robotics and/or control.
  - In a similar vain, I would not call the considered tasks 'robotic'. Some of them are more on the side of toy benchmark tasks, others are indeed robotics-inspired but still quite far from being representative for real robot tasks (see reasons above, and you already hinted at problems of Robosuit yourself [being essentially purely kinematic]).
  - On the flip-side, the tackled problem really is only relevant for real robots. For other policy learning settings - simulated robots, or maybe entirely different domains - the usefulness seems rather limited. Also, how much of a need for parsimonious sensing there really is, is also a bit debatable - it highly depends on the robot embodiment (light-weight UAV vs. stationary robot arm vs. networked swarm).
- The paper ignores a huge body of work. Work in the control community, in particular on event-based/event-triggered control/state-estimation, has studied this kind of problems since decades$^*$ and as a consequence come to the same insights. Overall the paper feels a bit like re-inventing the wheel (also e.g. the state-disagreement metric). Work on event-based sensing / event-based cameras also goes in a similar direction.
- With the above background, this paper serves as a nice confirmation about what common knowledge in control and robotics would predict.
  - When having the 'perfect world' assumption in mind, none of the findings are surprising, let alone counter-intuitive. In a perfect world, with perfect models, sensing etc. all tasks can be solved open-loop. Feedback is required when the models no longer hold (external perturbations, stochasticity, imperfect models) which you nicely show.
  - And then we can also think about making things more robust, i.e., finding a strategy that can deal with imperfect models/sensing, usually at the cost of performance, e.g., sim2real approaches.
- At least for the RL setting, I find the approach of using a policy that was trained closed-loop problematic as you point out in l. 339. It sounds a bit like "we got lucky". When instead training the policy with the parsimonious sensing, the policy will potentially change drastically to account for that. So the analysis is at best a proxy of what could be achieved under that setting.
- Minor
  - It would have been nice to promise releasing the code
  - typo l. 313 "hase"
  - typo l. 508 "efficient efficient"


$^*$ See e.g.

Trimpe, S. and D'Andrea, R., 2011. An experimental demonstration of a distributed and event-based state estimation algorithm. IFAC Proceedings Volumes, 44(1), pp.8811-8818.

For a cool real-world demonstration

**Questions:**

- I'd be really curious to see how the results change when introducing stochasticity/noise. I'd assume the VoSI profiles will change quite a bit, e.g. for the upright phase in swing-up
- Another interesting aspect would be to analyze the influence of the sensing-frequency on generalization. E.g. is more frequent sensing more robust to changes in the environment, perturbation, etc.?
- To push it even further you could consider analyzing what happens if you train the base-policies with domain randomization to make them more robust.

## After rebuttal
Thanks for the detailed and constructive replies. I have increased my score (as indicated in my message below).

---

> ### Author Response · Authors · 2024-11-20
>
> Thank you for such a detailed review! We provide responses to all your concerns below.
>
> **The paper studies entirely the wrong setting [for robots]… only an academic exercise with no practical use …  tasks are not “robotic” … robots have imperfect sensing / stochastic dynamics …**
>
> We ask that you evaluate this paper not as introducing an immediately applicable engineering solution, but as the first empirical approach to address a core scientific question in robotics: “How critical is sensory information at each moment to a pre-trained policy?” See our [global response](https://openreview.net/forum?id=ikr5XomWHS&noteId=Y159uHUuRZ) above for further discussion on the relevance of our findings to real robotic tasks.
>
> **The findings within the “perfect world” assumption are unsurprising: in a perfect world, with perfect models, sensing etc. all tasks can be solved open-loop.**
>
> We agree that this is indeed not surprising, this is why we explain this concisely in Lines 243-258, and use that as motivation to study settings with _imperfect_ models (or policies), due to limited policy capacity (see Ln 263 onwards). As such, almost none of our tasks can be truly solved open-loop, and the findings are only non-trivial because of this condition. To recap, our key findings are: In most tasks sensing only improves agent actions significantly at some rare moments during task execution, the reliance on environment feedback is similar for high-performing policies synthesized with different architectures, and the policy performance is inversely correlated with sensor dependence.
> Finally, note again that our experiments in this paper largely only study one axis of deviation (imperfect models) from the “perfect world (deterministic dynamics), perfect models, perfect sensing” base case, because these experiments are expensive to run, and we had to pick our battles. Do note in our global response we propose to study “imperfect sensing” in one task, to show that VoSI can be used to answer other questions beyond what we have demonstrated in this paper.
>
> **The tackled problem is only really relevant for real robots, usefulness limited in other domains. Even for robots, how much of a need for parsimonious sensing there really is, is debatable**
>
> We disagree. While we ourselves are motivated most by robotics applications, sensing often carries even more explicit costs in non-robotic decision-making settings. For example, when a government makes policy decisions about how best to serve its populace, it must perform a variety of expensive and time-consuming studies (such as censuses and surveys) to obtain “sensory observations” that determine its optimal policy decisions. Indeed, questions about the value of sensed information are studied outside of robotics too: for example, see our extended related work discussion section in Appendix F.
>
> Coming to robots, parsimonious sensing is often of value because it entails a series of other resource costs (not only communication, but also computation, time, energy, …) in the robot, and robots are often resource-scarce, as dictated by size, weight, and power constraints. As the reviewer themselves points out, the “event-triggered control” community, as well as many of the works pointed out in Related Work Sec 2, study this question for this reason.
>
> And finally, while we highlight the utility of VoSI characterization for efficient sensing, this is not its only use case. One might even view our approach as an extension to control of a kind of “interpretability” or “attribution” studies in other disciplines like computer vision and language. Just like they ask: “which parts of the image led the neural network to think that this is a cat / a spam email?” [1,2], one might use VoSI to ask “which sensory inputs most influenced the robot’s task performance?”
>
> **In the RL setting, using a policy that was trained closed-loop is problematic as you point out in l. 339. When instead training the policy with the parsimonious sensing, the policy will potentially change drastically to account for that. So the analysis is at best a proxy of what could be achieved under that setting.**
>
> Sec 2 lines 69-77 point out how we are different from methods that aim to generate policies under sensing constraints. We aim to characterize the value of sensory information for any learned controller, such that it might be relevant to executing the controller efficiently under any resource cost models or constraints. As such, we _are not aiming to generate new behaviors that permit less sensing_, just reproduce existing ones. Our proposed method reveals interesting insights into the properties of performant policies on various challenging robotic tasks of interest and serves as a general tool to understand the sensory requirements of any learned controller, complementing approaches that aim primarily to synthesize more efficient controllers.

---

> > ### Author Response · Authors · 2024-11-20
> >
> > **The paper ignores a huge body of work. Work in the control community, in particular on event-based/event-triggered control/state-estimation, has studied this kind of problems since decades and as a consequence come to the same insights. Overall the paper feels a bit like re-inventing the wheel (also e.g. the state-disagreement metric). Work on event-based sensing / event-based cameras also goes in a similar direction.**
> >
> > Thank you for pointing out the connection to event-triggered control. There are many bodies of work splintered across many disciplines asking questions relevant to this work, and we have tried to cover them in Sec 2 + our extended related work section in the Appendix F, but we agree that this is indeed a notable omission and have corrected it. The event-triggered control literature, much like the works discussed in Lines 72-76 aim to generate controllers that operate with limited sensing. Instead, we aim to _quantify the task value of sensory information to a fixed controller_. We are not aware of any works in the event-triggered control literature that do this.
> >
> > Perhaps more directly related to event-triggered control is our initial exploration of VoSI measurements to generate sensing-efficient executors of policies (Lines 510-519). However, event-triggered control typically (1) assumes that a low-cost sensor is constantly monitoring the state at each time $t$ and (2) defines a “trigger” event when a manually defined function of this sensory observation crosses a threshold. For example, in Trimpe and D’Andrea 2011, a trigger is defined based on the deviation of the coarsely sensed state from a model’s predictions, which is not a task-specific metric — for example, in the cup-catch task, deviations of the ball as it bounces around within the cup are largely irrelevant to task performance. _In contrast, our simple “efficient executor” (Lines 510-519) does not assume continuous monitoring or manually defined task-agnostic trigger events._
> >
> > Finally, as noted in the global response and in our response to Reviewer Tq4f above, we are now attempting to implement an event-triggered control inspired approach to serve as a useful (but in our opinion, non-critical) point of comparison.
> >
> > **It only becomes clear rather late in the paper that all experiments (except 4-room) are done in a setting assuming perfect sensing and that everything is deterministic.**
> >
> > We do say this early in the paper: Line 46 on page 1, but we agree that this is worth re-emphasizing to avoid inadvertently misleading the reader, and will do so. We also already discuss this limitation in Sec 6 too.
> >
> >
> > Thank you once more for investing your time and energy to help us to improve our paper. We believe that we have addressed your main concerns, but please do let us know if any concerns remain.
> >
> > **References**
> >
> > [1] Selvaraju, Ramprasaath R., et al. "Grad-cam: Visual explanations from deep networks via gradient-based localization." Proceedings of the IEEE international conference on computer vision. 2017.
> >
> > [2] Scott, M., and Lee Su-In. "A unified approach to interpreting model predictions." Advances in neural information processing systems 30 (2017): 4765-4774.

---

> > > ### Comment · Reviewer_6eFu · 2024-11-24
> > > **thanks for the detailed replies!**
> > >
> > > I appreciate the detailed replies, but am afraid it mainly comes down to question of perspectives.
> > > While the claims are now toned down in the replies, they are still in the title of the paper and e.g. also the "curiously correlated" in the abstract.
> > >
> > > I still believe the method itself is interesting (but then ideally it should in some way be compared to other analysis tools), but the replies to the question “How critical is sensory information at each moment to a pre-trained policy?” that you currently get through the experiments in the paper (that you also claim as contribution of the paper) are not - to put it in your own words: I'm still not convinced you picked the right battle. The proposed experiments are a nice step, though.
> > >
> > > I am still not convinced that studying converged policies in isolation is very valuable. The properties your method uncovers are tightly entangled with properties of the system (stochastic dynamics, stochastic sensing, etc.) and how the policy was obtained. The setting you tackle in the last part of the paper with 'sparsifying' an existing policy still seems marginal - like you wrote, if you really care about limited resources you'd train the model differently to start with.
> > >
> > > And finally regarding the tasks: Yes, those are common in the robot learning community. They are fine as standardized benchmarks for some things but many people will agree that they have only very limited predictability on how well something will work in a practical, real robot task. Compared to some other robot learning papers, the task you tackle is a lot more closely tied to the deviations form the 'perfect world' assumptions - which in my opinion poses a lot more stringent demands on the tasks compared to studying some other properties, and the current tasks don't seem to be a good fit for that.
> > >
> > > I really appreciate all the additional materials that now got added to the appendix. It's going to be a struggle to squeeze all that in the main paper as promised...
> > >
> > > To sum up, while the replies addressed some of my concerns I'm still not convinced by the tasks, the studied setting, and the associated lessons learned.
> > >
> > > My new score would be a "4" (which does not exist). It would be great to see the results of the pending experiments and the updated things properly integrated in the main paper.

---

> > > > ### Author Response · Authors · 2024-11-27
> > > >
> > > > We thank the reviewer for their response. We have now incorporated the promised changes and results of the experiments proposed into the main paper as outlined in the [global response](https://openreview.net/forum?id=ikr5XomWHS&noteId=rvYibL0A1s). We summarize the key updates below:
> > > >
> > > >
> > > > - As promised, we have now performed an experiment comparing our proposed greedy strategy with an event-triggered control approach that has extra access to a coarse sensor. We report the results in Appendix H. Note that event-triggered sensing operates under more relaxed assumptions than us: instead of completely foregoing sensing at some moments in time, they assume continuous access to coarsened sensory inputs: in our implementation, this corresponds to always knowing which room the agent is in (“room sensor”).  This extra knowledge proves to be a significant advantage in the deterministic dynamics setting (Fig 18, left), but when dynamics is stochastic, our naive VoSI-based greedy strategy matches this approach in performance, even without a room sensor.
> > > > - We report the results of our analysis of VoSI for an agent operating with noisy-sensing on the cartpole swingup task in Appendix I. We train a small CNN-based visual state estimator that has up to 5 cm and 2 degrees error, which is quite significant in this task. As such, the value of new sensory measurements increases: this is reflected in steeper VoSI profiles overall. However, trends remain similar: for example, we see stepped VoSI profiles during the swingup phase, and close to flat VoSI profiles even up to 70 steps of open-loop execution during the balancing phase when the pole is already close to vertical. This establishes both that our VoSI framework is applicable to many more general task setups than reported in the main paper, and that the findings reported in the main paper already offer useful intuitions for what results in more general/complex task setups might look like.
> > > >
> > > > > **I still believe the method itself is interesting (but then ideally it should in some way be compared to other analysis tools)**
> > > >
> > > > We thank the reviewer for the comment. As explained in Sec 2 related work, there aren’t any easily comparable methods in the prior literature to our approach for quantifying the value of sensory information. However, we welcome any specific suggestion here, and will try to incorporate them.

---

> > > > > ### Comment · Reviewer_6eFu · 2024-11-27
> > > > >
> > > > > Thanks again for the replies.
> > > > >
> > > > > The additional experiments/results are indeed insightful and it's a bit a pity that those now got relegated to appendices. I'm still not convinced that the results in the main paper provide the most useful intuitions, also compared to the newly added things - which would have required a more significant modification of the main paper.
> > > > >
> > > > > I'm also unfortunately not aware of any easily comparable methods.
> > > > >
> > > > > I'll increase my score.

---

> > > > > > ### Author Response · Authors · 2024-12-01
> > > > > >
> > > > > > Thank you, we believe that the new results on event-triggered control and noisy sensing swingup can very much be incorporated into the paper’s flow: they are already very much in line with the narrative of the paper, and it is only a matter of space. For our next revision, we may plan to condense our discussion in Sec 4.1 to be less elaborate (and move the current version of Lines 210-260 to appendix).
> > > > > >
> > > > > > Thank you for all your efforts towards helping us improve our paper already, and do let us know if you have any further concerns!
> > > > > >
> > > > > > Finally, the lack of comparable methods is indeed why we are excited to work in this direction: the questions we ask in this work are very simple ones to ask about robotics and decision-making more broadly, but have proven to be very hard to answer over decades, and not for lack of trying (our extended related work in Appendix F spans work across robotics, control theory, machine learning, and economics)! We hope that our work will be a catalyst (and a straightforward data point for comparisons) for a new spell of vigorous efforts in this direction!

---

### Official Review · Reviewer_Tq4f · 2024-11-08

**Soundness:** 3
**Presentation:** 4
**Contribution:** 4
**Rating:** 8
**Confidence:** 3

**Summary:**

This paper presents a method to answer the question in the title. More specifically they approach the problem as how does the frequency of observation updates change policy performance. They perform many studies on various environments to answer this question.

**Strengths:**

Research Problem
Very good. In my view there are two research questions that this work attempts to answer. One is “how important is recent information to learned policies?”, the other is “what is the best way to determine how important is recent information to learned policies?”. Both questions are interesting and relevant to the field.

Novelty
Very good. I’m not very familiar with the sensing in agent learning literature but if their related work is complete (which it seems to be), their work has significant novelty.

Significance/Contribution
Very good. This work could have a lot of significance in the field. The idea of not only discovering the importance of information to policies (and robots) but also the general idea of how to study this question could provide many other works a base to work off. My one complaint is that the introduction doesn’t give the authors perspective on their exact contribution with this work. Adding a few lines in with their contribution would be great as someone who is more broadly in the RL field but not the specific questions of sensing for RL and learning.

Algorithm Details
Good. There isn’t an algorithm in this work exactly, but their method is described well. It isn’t an extremely complex idea (using policies that can predict multiple actions in the future) but from their description I believe I could recreate their work.

Experimental Selection
Excellent. Most of this paper is made up of experiments and analysis to answer their questions. I think the questions presented are well thought out.

Presentation of Results
Good. In general, the results are clear and the figures well done.

Clarity
Very good. In general, this paper is very well written and clear. I had no issues reading or understanding the ideas.

Future Work/Conclusion
Very good. I think the future work presented in this paper answers the ideas I had for future work. I think the authors can build upon it in interesting ways and can continue to grow our understanding of these questions presented in the paper.

**Weaknesses:**

The one figure I’m still confused about is Figure 9. I don’t completely understand the figure even with the description in the text.

Statistical Rigor
I would like confidence intervals on all of the figures. They are on some but figures 3, 4, 5 and 11 don’t have them. Having CIs is critical to understanding the data.

**Questions:**

Baseline Comparisons
There are no baseline comparisons in this work. It seems to be that there is no comparable baseline to have here from the related work. That is reasonable to me but I wonder if for the results presented in figure 11 you might be able to have a baseline comparison? The idea is for different amounts of maximum regret you have some amount of sensory interaction. Is it possible to do some comparison here for the other works that also reduce sensory interaction?

---

> ### Author Response · Authors · 2024-11-20
>
> Thank you for the kind words and feedback!
>
> **“Adding a few lines in the introduction on their contributions?”**
>
> Thank you, we have edited the manuscript to reflect the contributions clearly in the introduction. Further, we have provided a summary of our contributions in the [global response](https://openreview.net/forum?id=ikr5XomWHS&noteId=Y159uHUuRZ) above.
>
> **“Figure 9 is confusing”**
>
> Thank you, we did indeed find this figure particularly difficult to explain well within limited space, and will try to improve now. As you will recall, Figs 7 and 8 show VoSI(s,h) for a single state s while varying the horizon h. In Fig 9, we wanted to simultaneously vary not just the open-loop horizon h (from 1 to 100) but also the state s (from starting state $s_0$ to ending state $s_T$ of a closed loop policy rollout).
>
> - This means we are now mapping a scalar function VoSI($s_{t_1}$, $h$) of two arguments (time index $t_1$ of last-observed state $s_{t_1}$, and open-loop horizon $h$). **We use colors to indicate the VoSI values**, and the two arguments are on the x and y axis.
> - In particular, **the x-axis** of the figure represents the time index $t_1$ of the last-observed state $s_{t_1}$. **The y-axis** represents the _offset_ open-loop horizon $t_1+h$.
> - Each vertical column of colors corresponds to the VoSI profile for a particular state at varied horizons: this is a 1-D representation of the same kind of VoSI profiles depicted in Figs 7 and 8. To draw the VoSI profile for a particular last-observed state, say, $s_{t_1=10}$, one could look up the column for $t_1=10$. For example, for $h=1$, we can look up the color for $y=t_1+h=10+1=11$, and match it to the color chart on the right.
> - The choice of adding the last-observed time offset to the y-axis i.e. using $t_1+h$ rather than just $h$ helps to highlight the structure in the evolution of VoSI over time. Tracking along a horizontal row, say $t_1+h=40$, you can read off how important re-sensing before timestep 40 was, at various prior moments in time: i.e. it was very important to re-sense if $t<20$ i.e., the agent had not yet observed t=20, corresponding to “high” colors. However, for $t>20$, there was no need to re-sense before $t+h$ = 40, corresponding to “low” colors.
> - With this interpretation, we can now observe from Fig 9 that for all last-observed states $t_1<20$, VoSI profiles are consistent: they are low up to $t_2=20$ (no task-relevant information from sensing before timestep 20), and high afterwards (suggesting critical information at timestep 20). Likewise, VoSI profiles are also consistent for all last-observed states $t_1>20$: there is no more value to sensing any more, for ever after.
> - Upon inspection, we find that timestep 20 in the closed-loop trajectory corresponds to a tipping point after which the ball is destined to fall into the cup (near the lip and with appropriate velocity), which means the task will be complete.
>
> **Statistical Rigor and plots with confidence interval(CI)s**
>
> Operating within space constraints, we dropped the confidence intervals to maintain legibility while making our plots smaller. We have now added larger plots with CIs in Appendix G (Figures 17-20). The trends pointed out in the paper remain robust.

---

> > ### Author Response · Authors · 2024-11-20
> >
> > **No baselines for the rest, but could be for Fig 11?**
> >
> > Thank you, yes, there is indeed no baseline we are aware of that assigns a task value to sensing information, comparable directly to our VoSI proposal.
> >
> > It is also true that Fig 11 might permit some comparison to prior relevant work at least in theory. To recall, Fig 11 shows proof-of-concept for a potential application of VoSI to reduce sensing needs when executing a pre-trained policy, using a simple greedy approach that can adapt to test-time-specified sensing costs. This is a rather simplistic implementation of how VoSI might be useful in this application, but we agree that it may be worth exploring prior methods for reduced-sensing during control in this context. Unfortunately, we have found no reasonable implementations of such approaches that can be applied directly to any of our tasks, but we are attempting to implement some by ourselves:
> > - **(A)** We have now unsuccessfully attempted to follow the recipe of (Hansen,1996) (cited in Sec 2 first paragraph), which trains RL policies for a pre-specified (rather than specified at test time) sensing cost model. Even in our simplest grid-world tasks, this proved too costly to run: the method relies on searching over open loop action sequences, which grows exponentially with the admissible time horizon.
> > - **(B)** [ongoing attempt] Finally, reviewer 6eFu, points out the literature on  “event-triggered control”, which is also relevant here — it does not involve completely forgoing sensing at any moment but instead choosing to sense at coarser fidelity until “an event is triggered” (see our response to 6eFu for more details). For example, in the cartpole case we can define event-triggers based on a coarse measurement of the pole angle; when the measurement breaches a manually defined threshold (e.g. >5 degrees from vertical), then higher-fidelity sensing kicks in, and the lookahead policy is queried again for a new plan. We plan to implement such a baseline on the four-rooms task by defining events that operate on coarsely sensed transitions between sub-grids and contrast it with our proposed method based on VoSI profiles.
> >
> > Note that “baseline” B relies on manually defined heuristics, which in our case might be informed by VoSI profiles. As such, we do not see this as a competitor for our VoSI-based sensing reduction proto-implementation, but rather as an alternative to the naive greedy strategy implemented there.
> >
> > Finally, while we are attempting to implement this comparison, we do not see them as central to the paper, which is mainly about quantifying the value of sensory information in complex tasks with state-of-the-art policies. The reduced sensing application in the paper is only a proof of concept, and one example of the ways in which VoSI might become practically useful.

---

> > > ### Comment · Reviewer_Tq4f · 2024-11-22
> > >
> > > For the baseline question. That is a reasonable response to my question. I agree it isn't central to the paper and don't believe no baseline hurts it. If you can get a baseline working in time for publication that could only improve it in my mind but isn't required.

---

> > > > ### Author Response · Authors · 2024-11-27
> > > >
> > > > We thank the reviewer for their response. We have now incorporated the promised changes and results of the experiments proposed into the main paper as outlined in the [global response](https://openreview.net/forum?id=ikr5XomWHS&noteId=rvYibL0A1s). We summarize the key updates below:
> > > >
> > > > - We have now revised the regret profile plots in the main paper to incorporate 95\% CIs.
> > > > - Additionally, we thank the reviewer for their clarification question on Figure 9. We have made small changes to the notation and axis labels in the main paper surrounding Figure 9 for clarity, and have incorporated a summary of our response here in Appendix E for future readers.
> > > > - As promised, we have now performed an experiment comparing our proposed greedy strategy with an event-triggered control approach that has extra access to a coarse sensor. We report the results in Appendix H. Note that event-triggered sensing operates under more relaxed assumptions than us: instead of completely foregoing sensing at some moments in time, they assume continuous access to coarsened sensory inputs: in our implementation, this corresponds to always knowing which room the agent is in (“room sensor”). This extra knowledge proves to be a significant advantage in the deterministic dynamics setting (Fig 18, left), but when dynamics is stochastic, our naive VoSI-based greedy strategy matches this approach in performance, even without a room sensor.

---

> > ### Comment · Reviewer_Tq4f · 2024-11-22
> >
> > Contributions addition helps a lot thanks for adding that.
> >
> > Figure 9 makes sense now with more explanation. I'm not sure if it was simply my not understanding or if you could improve the figure/text some to make it better. I'll think about it and update this if I can think of a way to do so.
> >
> > I understand wanting to make them clear. I personally like the figures 17-20 more than the ones in the main text because of the CIs. In my opinion it isn't more difficult to read and conveys more information.

---

### Author Response · Authors · 2024-11-20
**Global Response**

We thank all the reviewers for their time and detailed comments. We are excited that all the reviewers found the research questions posed interesting and relevant and also found our proposed methodology sound and experiments well executed.

We would like to use this global comment to address a few common questions, highlight key changes to the paper responding to the reviews, and share planned experiments:

**Clarification on the contributions:**

We have modified the introduction of the paper to include a few lines summarizing the following key contributions of this work:
1. Our work is the first to propose **an approach to quantify the value of sensory information (VoSI)** in complex task setups that are commonly considered in benchmarks.
2. With VoSI as a probing tool, we have conducted a first experimental study across **diverse tasks and state-of-the-art learned policies that are more complex than prior works in this space**. Our results align well with what one might intuitively expect in simple settings, and further, in more complex settings where such intuitions are difficult a priori, they produce novel insights about the value of sensory information to task policies. For instance, even on more complex tasks like cartpole-swingup, cup-catch, and Push-T sensing only improves a learned lookahead policy’s actions significantly only at rare moments during task execution, affording at least a 5 times reduction in sensing required.
3. Experimental analysis is expensive, and in this first effort, we have largely explored imperfect models in environments with perfect sensing and deterministic dynamics, but **VoSI is applicable to other practical settings too, where we have only established proof of concept (stochastic dynamics, sensor/model noise)** and much more remains to be explored. What we have presented here is, we hope, only the tip of the iceberg, and already exciting and informative to the research community.
4. Furthermore, we have highlighted the **potential of VoSI for efficient policy execution** by demonstrating a proof-of-concept greedy strategy that achieves better performance than simplistic fixed-rate sensing policies.

**[Reviewer 6eFu, Reqd]: On the applicability of the study to real robotic settings with imperfect sensing and stochastic dynamics.**
```
Reviewer 6eFu:  “The paper studies entirely the wrong setting [for robots]… only an academic exercise with no practical use …  tasks are not “robotic” … robots have imperfect sensing / stochastic dynamics … “
```
```
Reviewer Reqd: “The paper aims to provide insights for robotics tasks, yet all results are obtained only in simulation. The characteristics of simulated benchmarks are clearly very different from real-world robotics tasks..”
```
We request you to evaluate this paper not as presenting a new engineering artifact immediately applicable to robots today, but instead as the first to present an empirical approach for **answering a core scientific question in robotics: how important (measured in terms of task returns) is sensory information at each moment in time to a (state-of-the-art) pre-trained policy?** Framed another way, what is the extent to which each sensory measurement influences the task performance of a given policy? As all reviewers agree:  “this is an interesting and understudied problem, and the method for analysis is sound.”


Questions like these have been raised and studied for decades (see Lines 31-45, our Related Work Sec 2 Lines 67-102), but typically in stylized tasks and with simplistic policies that are very far from the kinds of tasks that are today of broad interest in robotics, particularly robot learning. For example, Majumdar et. al (2023)[1] study tasks with a few discrete actions (or continuous action primitives) and a very short horizon to obtain computationally tractable bounds on performance of a sensor system, Erdmann (1995)[2] characterize requirements on sensors from an analytical and geometric understanding of stylized planar tasks like point to disk, screw in plate, etc. under specific assumptions on the policy class. **The tasks and policies that we study in this work (see Fig 2) are already significantly more complex and much closer to practical utility than the most closely related prior studies**. Further, there is reason to believe that our insights in these settings carry over to real robots: for example, recent literature[3,4] show that open loop baselines are quite effective in real world manipulation and locomotion tasks (Lines 191-193). Finally, note that all tasks studied in this work are common simulated benchmarks that are often used in the robot learning community [e.g. 5-7], and one note of caution from our paper is that these tasks might not sufficiently test sensing / state estimation / perception capabilities.

---

> ### Author Response · Authors · 2024-11-20
> **Global Response (cont.)**
>
> As with any new scientific directions, **all the practical use cases will only be established through many follow-up efforts** (including our own that are already in progress), and our aim in this paper is mainly to establish VoSI measurements as a new probing tool capable of offering insights to researchers and practitioners. **The experiments conducted in the paper are at once (1) extensive** (they span more tasks and more state-of-the-art policy classes than in any prior related works, and even just the plots reported in the paper now required over 600 A40 GPU hours of compute), and **(2) still limited relative to all potential uses for VoSI**. In particular, we largely only explore one axis of deviation from the trivial “deterministic world, perfect models, perfect sensing” base case (where no sensing is required): where the models are imperfect.  However, we fully expect that **VoSI is also applicable in sensing-limited and stochastic-dynamics settings**. As proof of concept, we had already applied such a noisy dynamics setup in the simple gridworld task in Sec 4.1. We had also argued there that limited policy expressivity for all practical purposes is equivalent to such noise, which is why we need to still sense the environment at many instances within our studies.  We acknowledge these limitations already in Sec 6 in the paper.
>
> **(in progress) A new experiment with sensing noise**: Finally, to further demonstrate that our analysis can be extended to settings with realistic sensing noise, we plan to report experiments with a policy that has a CNN-based visual state estimator operating from image observations for the cartpole swingup task.
>
> **Key Changes in the Paper:**
> * We have added a more thorough extended related works section (Appendix F) to discuss the adjacent literature.
> * We also present in Appendix G our initial efforts in addressing reviewers’ suggestions and also include placeholder sections with the planned experiments that we are currently conducting.
>
> **Planned Experiments:**
> 1. In response to the suggestion from reviewer Reqd, we redid the grid world experiments, corresponding to the how dynamics error necessitates more sensing (Figure 4 (right)) and the demonstration of the performance of a greedy-strategy informed by our analysis (Figure 11) , with an error model that sometimes moves the agent to a wrong but neighboring cell, rather than teleport it to any cell over the full grid. We show our results in Appendix G.2 in Figure 19 and Figure 20. All the performance trends reported in the paper continue to hold. We will move this into the main paper with suitable changes.
> 2. In response to the suggestion from reviewer 6eFu, we propose to conduct a proof-of-concept study demonstrating the applicability of VoSI in environments with realistic sensing noise. We plan to replicate our analysis with a policy that uses a CNN-based visual state estimator operating from image observations on the cartpole swingup task.
> 3. In response to the suggestion from reviewer Tq4f, we propose to characterize an event-triggered control inspired baseline that reduces sensing for control beyond the demonstrated simplistic greedy strategy leveraging VoSI profiles. We plan to define heuristic event-triggers based on course measurements of the state (e.g. a sensor that detects changes in sub-grids the agent is in) to define when high-fidelity state measurement is made to query for a new plan. However, note that different from our proposed method, this baseline relies on some amount of coarse sensory measurement at every timestep to trigger events.
>
> **References:**
>
> [1] Majumdar, Anirudha, Zhiting Mei, and Vincent Pacelli. "Fundamental limits for sensor-based robot control." The International Journal of Robotics Research 42.12 (2023): 1051-1069.
>
> [2] Erdmann, Michael. "Understanding action and sensing by designing action-based sensors." The International journal of robotics research 14.5 (1995): 483-509.
>
> [3] Dasari, Sudeep, et al. "RB2: Robotic manipulation benchmarking with a twist." arXiv preprint arXiv:2203.08098 (2022).
>
> [4] Raffin, Antonin, et al. "A Simple Open-Loop Baseline for Reinforcement Learning Locomotion Tasks." arXiv preprint arXiv:2310.05808 (2023).
>
> [5] Singh, Sumeet, et al. "Multiscale sensor fusion and continuous control with neural CDEs." 2022 IEEE/RSJ International Conference on Intelligent Robots and Systems (IROS). IEEE, 2022.
>
> [6] Mandlekar, Ajay, et al. "Mimicgen: A data generation system for scalable robot learning using human demonstrations." arXiv preprint arXiv:2310.17596 (2023).
>
> [7] Nasiriany, Soroush, et al. "RoboCasa: Large-Scale Simulation of Everyday Tasks for Generalist Robots." arXiv preprint arXiv:2406.02523 (2024).

---

> > ### Author Response · Authors · 2024-11-27
> > **Update on the Global Response (Dated Nov 27)**
> >
> > We thank the reviewers for their continued feedback. As promised in our earlier round of responses from Nov 19, we have now moved as much of the new material from the appendices into the main paper as space permits, and added summaries and pointers to appendices when this was not possible.  We have also performed all the experiments we had promised, and the results are now incorporated into the manuscript. The key changes to paper and the summary of the key findings from the experiments conducted during the rebuttal phase is listed below:
> >
> > - In response to reviewer Reqd’s suggestion we now incorporate a more local error model for dynamics in the grid world experiments and have updated Figure 4 and Figure 11 in the main paper to reflect this change. We have also suitably updated the description of the procedure used to obtain the inaccurate models used in our experiments in Appendix B. The trends are consistent with what we had previously reported, as Reqd acknowledged in their response.
> > - In response to a suggestion from reviewer Tq4f, we have now implemented an “event-triggered control”-based approach on the four-rooms task, to offer a point of comparison for the VoSI-based greedy efficient execution strategy presented at the end of Sec 5 (Fig 11). We report the results in Appendix H. Note that event-triggered sensing operates under more relaxed assumptions than us: instead of completely foregoing sensing at some moments in time, they assume continuous access to coarsened sensory inputs: in our implementation, this corresponds to always knowing which room the agent is in (“room sensor”).  This extra knowledge proves to be a significant advantage in the deterministic dynamics setting (Fig 18, left), but when dynamics is stochastic, our naive VoSI-based greedy strategy matches this approach in performance, even without a room sensor.
> > - In response to the suggestion from reviewer 6eFu, we present the results of our analysis of VoSI for an agent operating with noisy-sensing on the cartpole swingup task in Appendix I. We train a small CNN-based visual state estimator that has up to 5 cm and 2 degrees error, which is quite significant in this task. As such, the value of new sensory measurements increases: this is reflected in steeper VoSI profiles overall. However, trends remain similar: for example, we see stepped VoSI profiles during the swingup phase, and close to flat VoSI profiles even up to 70 steps of open-loop execution during the balancing phase when the pole is already close to vertical. This establishes both that our VoSI framework is applicable to many more general task setups than reported in the main paper, and that the findings reported in the main paper already offer useful intuitions for what results in more general/complex task setups might look like.
> > - In response to reviewer Tq4f’s suggestion we now incorporate 95% confidence intervals in all the plotted task-regret profiles in the main paper.

---

### Meta-Review · Area_Chair_HBb6 · 2024-12-20

**Metareview:**

This paper investigates the value of sensory information in robotic tasks by analyzing how withholding observations affects performance across various policy synthesis methods in simulated tasks. It addresses a novel and important question, offering insights that could inform more efficient decision-making in robotics.

Reviewers agree the problem is interesting and unexplored, with reviewer 6eFu suggesting reframing and citing other relevant literature for greater impact, while other reviewers see value in its current form as a foundation for future work.

I recommend acceptance of this paper but urge the authors to incorporate feedback from reviewers in the final version to clarify the limitations of this work in text.

**Additional Comments On Reviewer Discussion:**

The authors clarified confusions and details regarding the problem setting and experiments during discussion. The reviewers actively participated in the discussion and coordinated throughout.

---

### Decision · Program_Chairs · 2025-01-22

Accept (Poster)